# Soft ferroelectret ultrasound receiver for targeted peripheral neuromodulation

Tong Li[1,2,3,7], Zhidong Wei [1,7], Fei Jin [1,7], Yongjiu Yuan[2], Weiying Zheng[1], Lili Qian [1], Hongbo Wang [2], Lisha Hua[2,4], Juan Ma[1], Huanhuan Zhang[2], Huaduo Gu[2], Michael G. Irwin [4], Ting Wang [5] ✉, Steven Wang[2,3] ✉, Zuankai Wang [6] ✉ & Zhang-Qi Feng [1] ✉

Bioelectronic medicine is a rapidly growing field where targeted electrical signals can act as an adjunct or alternative to drugs to treat neurological disorders and diseases via stimulating the peripheral nervous system on demand. However, current existing strategies are limited by external battery requirements, and the injury and inflammation caused by the mechanical mismatch between rigid electrodes and soft nerves. Here we report a wireless, leadless, and battery-free ferroelectret implant, termed NeuroRing, that wraps around the target peripheral nerve and demonstrates high mechanical conformability to dynamic motion nerve tissue. As-fabricated NeuroRing can act as an ultrasound receiver that converts ultrasound vibrations into electrostimulation pulses, thus stimulating the targeted peripheral nerve on demand. This capability is demonstrated by the precise modulation of the sacral splanchnic nerve to treat colitis, providing a framework for future bioelectronic medicines that offer an alternative to non-specific pharmacological approaches.

Action potentials in nerve tissue play a central role in cell–cell communication. They propagate signals along the neuron's axon toward synaptic boutons, which then connect with other neurons at synapses, or to motor cells or glands. In other types of cells, they activate intracellular processes, which, for example, can lead to muscle contraction or release of hormones such as insulin[1,2]. Virtually all organs and their functions are regulated through such nerve impulses. Consequently, electrical stimulation of the nervous system has attracted great attention for potential biomedical therapeutic applications[3–5]. In particular, the ability to precisely target peripheral nerves and, therefore, specific organs and tissues is emerging as an engaging concept for neuromodulation in many chronic diseases[6–9]. Traditionally, most of these treatments currently require external energy input and wired

connections to apply electrostimulation pulses to nerve tissues via implanted rigid metal electrodes. However, these battery-powered tethered systems can restrict natural motions and prevent social interactions[10–13]. Besides, a mechanical mismatch at the rigid electrode–soft neural tissue interface can cause trauma and insertion-related lesions, inflammation reactions, and even neuronal apoptosis, ultimately leading to therapeutic failure[10,14,15].

Recognizing this defect, numerous endeavors have been made to develop soft electrodes with mechanical properties close to neural tissue[16–20], or rigid highly miniaturized wireless stimulators[6,21,22]. Nevertheless, devices that use these designs still require external power supplies and connecting wires or rigid components, and thus may not be the ideal solution. Recently, the ultrasound (US) receiver

[1]School of Chemistry and Chemical Engineering, Nanjing University of Science and Technology, Nanjing 210094, China. [2]Department of Mechanical Engineering, City University of Hong Kong, Hong Kong 999077, China. [3]Research Center for Nature-inspired Engineering, City University of Hong Kong, Hong Kong 999077, China. [4]Department of Anaesthesiology, The University of Hong Kong, Hong Kong 999077, China. [5]State Key Laboratory of Bioelectronics, Southeast University, Nanjing 210096, China. [6]Department of Mechanical Engineering, The Hong Kong Polytechnic University, Hong Kong 999077, China. [7]These authors contributed equally: Tong Li, Zhidong Wei, Fei Jin. ✉e-mail: tingwang@seu.edu.cn; steven.wang@cityu.edu.hk; zk.wang@polyu.edu.hk; fengzhangqi1981@163.com

has been proposed as a wireless, leadless, and battery-free electrical stimulator as it can convert US waves into electrical power via its piezoelectric element[10,23–26]. Unfortunately, commonly used high-performance piezoelectric ceramics have brittle mechanical properties, comparatively high mass densities, and rigid and planar characteristics[27,28]. Although much effort has been attempted to overcome these limitations by developing piezoelectric composite materials that combine piezoelectric ceramics and polymers, they have been unsuccessful as either piezoelectricity or flexibility and processability are compromised. Polymeric piezoelectric materials,

represented by poly(vinylidene fluoride) (PVDF), offer superior flexibility and excellent biocompatibility, but their weak piezoelectricity largely limits their electric physiotherapy as effective US receivers, unless flexibility is sacrificed[29–31]. Herein, we utilize the cavitation effect to introduce ferroelectret pore structure into the soft electrospun PVDF fiber matrix, which can obtain additional electric dipoles at the surface of the material and inside it, thus enhancing the electrical performance output. As-fabricated fibrous ferroelectret non-woven fabric with low Young's modulus can act as a US receiver by wrapping target peripheral nerve (PN) tissue (upper inset of Fig. 1a). This

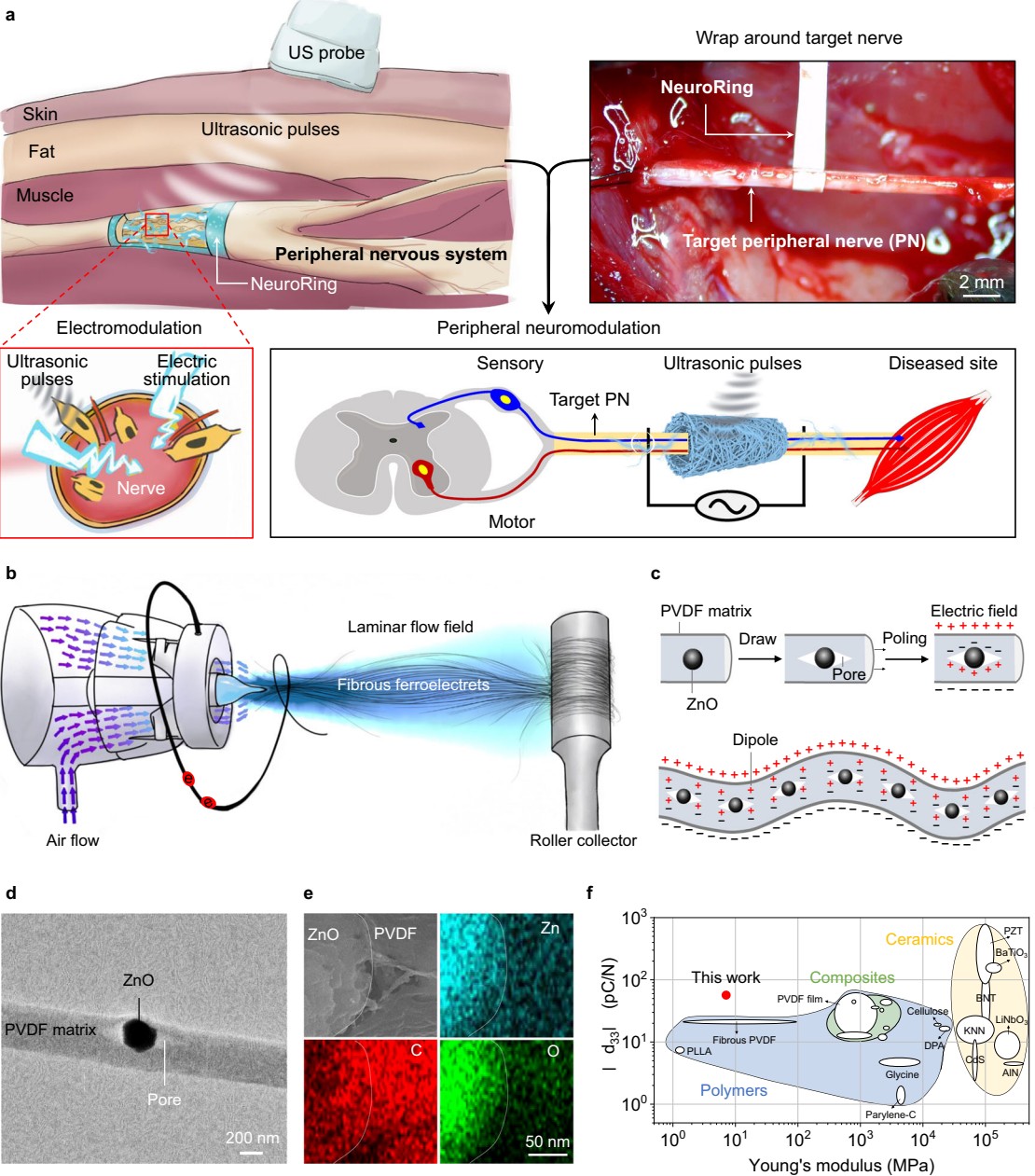

**Fig. 1 | Concept and material design of ferroelectret-based NeuroRing for peripheral neuromodulation. a** The NeuroRing wrapped around a peripheral nerve (PN) is an US receiver that can be triggered by US pulses to yield electrical impulses targeting the PNs that regulate various organs and functions. **b** Illustration of the laminar-flow-assisted electrospinning for fabricating ferroelectret fibers. **c** Ferroelectret was a composite consisting of piezoelectric PVDF and ZnO particles with a porous ferroelectret structure. The cavitation effect of the gas stretching force creates voids at the particle-polymer interface. These pores induce

electrically polarized domains (dipoles) in a subsequent process called poling. The increased numbers of dipoles improve the output of the electric charges that are formed by US vibrations. **d** TEM image showing the ferroelectret pore structure. $n = 8$ independent experiments showing similar results. **e** EDS mapping images of fiber localization. $n = 3$ independent experiments showing similar results. **f** Piezoelectric coefficient versus Young's modulus for various piezoelectric materials. Detailed data and references are in Supplementary Table 1.

NeuroRing can be activated by US vibrations to yield electric impulses that modulate the target nerve for disease intervention and treatment as necessary (lower inset of Fig. 1a).

## Results

The ferroelectret fiber was a composite of PVDF fibrous matrix and zinc oxide (ZnO) particles, fabricated using a laminar-flow-assisted electrospinning method (Fig. 1b). We introduced a coaxial air-blowing setup into the regular electrospinning device. High-velocity air blown out from the outer nozzle formed a jet that merged into a laminar-flow field behind the Taylor cone. This air-blowing force augmented the cavitation effect between PVDF and ZnO (Fig. 1c, Supplementary Fig. 1), as evidenced by the voids around the ZnO particles observed only in the drawn fibers (Fig. 1d, e, Supplementary Fig. 2). ZnO was uniformly distributed in the PVDF matrix as shown by elemental mapping (Supplementary Fig. 3). Also, this enhanced cavitation effect was universal and applicable to various sizes of ZnO and even to agglomerated particles (Supplementary Fig. 4). The fiber was stretched and polarized in a high electric field of 1.2 kV/cm, and then oriented macromolecular chains and their crystalline structures along the fiber axis as it formed (Supplementary Fig. 5), as well as induced the formation and alignment of $CH_2$-$CF_2$ dipoles in a high electric field of 1.2 kV/cm. Ultimately, more electric dipoles were generated at the interfaces between the polymer matrix, particles, and these elongated voids (Fig. 1c). This increased number of dipoles enhanced the spontaneous electric charges generated by US vibrations (Supplementary Movie 1). This explained the high piezoelectric coefficient $d_{33}$ observed only in the drawn ferroelectret fibers (Supplementary Fig. 6). The -$d_{33}$ we measured was as high as $56 \pm 2$ pC $N^{-1}$, considerably higher than that piezoelectric polymers as well as even comparable to piezoelectric ceramics (Supplementary Table 1).

The fibrous PVDF introduced with ZnO did not reduce its flexibility, and was able to tolerate deformations such as bending and twisting (Supplementary Figs. 7 and 8). The Young's modulus value of the fibrous ferroelectret non-woven fabric was estimated to be 7.6 MPa when measured via tensile tests (Supplementary Fig. 9), which was in the same order as seen with soft tissue, for example, nerve tissue[32]. This modulus value was four to six orders of magnitude lower than that of commonly used piezoelectric ceramics such as PZT and $BaTiO_3$, and its piezoelectric coefficient was nearly 1 order of magnitude higher than piezoelectric polymers with similar modulus value (Fig. 1f). Besides that, Young's modulus value of the single ferroelectret fiber was estimated to be 2.7 GPa when measured via nanoindentation experiments using an atomic force microscope (Supplementary Fig. 10). It approximated the range of various fibrous biopolymers such as natural piezoelectric collagen (0.5–10 GPa)[33]. As a result, our fibrous ferroelectret non-woven fabric was able to conformally wrap neural tissue, indicating high conformability to nerve tissue (Supplementary Fig. 11).

To detect the dynamic change in charge release, the fibrous ferroelectret non-woven fabric constituting the NeuroRing was submerged in ethanol at 10 mm under an US probe setup (left, Fig. 2a). The test device operated in single-electrode mode using an Al foil as the electrode, and a Cu wire conducting the charges (right, Fig. 2a). For safety, we adopted US with a frequency of 1 MHz and an intensity of 0.5 W $cm^{-2}$, commonly used in clinical US physiotherapy as the pulse parameter. Under these experimental conditions, the acoustic pressure applied to the ferroelectret device (dimension of 5 mm by 5 mm) was 117 kPa (Supplementary Fig. 12a). It yielded a peak-to-peak voltage (Vpp) of 6 V (Fig. 2b), significantly higher than the air-blowing PVDF fibers (Vpp of 1.6 V, Supplementary Fig. 12b) and PVDF/ZnO composite fibers (Vpp of 2.7 V, Supplementary Fig. 12c). The voltage fluctuation period was 1 μs, which was in accordance with the US frequency of 1 MHz. Besides, compared with the traditional foamed ferroelectrets that are prone to depolarize[34], our ferroelectrets showed excellent stability. During the 6-month period, the output Vpp was maintained at $6 \pm 1$ V with almost no change (Supplementary Fig. 13a). Thanks to the ring-shaped design that wraps around the nerves, our NeuroRing can reduce the impact of rotation misalignment between the ultrasound probe and the ultrasound receiver (Supplementary Fig. 13b), a common issue in ultrasound power transmission[10].

Next, due to its similarity to human skin in terms of anatomy and composition, a porcine tissue was used to evaluate the US-triggered transient performances of our non-woven fabric that constituted the NeuroRing. Considering that the purpose of our NeuroRing was to modulate the peripheral nervous system branching out from the brain and spinal cord (Supplementary Fig. 14), the distribution depth in human tissue ranged widely from a few millimeters to more than ten centimeters[35]. Therefore, we inserted the packaged non-woven fabrics (dimension of 5 mm by 10 mm) into porcine tissue to depths ranging from 10 mm to 10 cm, followed by ex vivo testing (Fig. 2c, Supplementary Fig. 15a). A commercial US gel was applied to the porcine skin to minimize loss of incident acoustic energy. At 10 mm, the fibrous ferroelectret non-woven fabric yielded a Vpp of 2.5 V (left, Fig. 2d). Owing to the increased acoustic impedance and attenuation in the different media and layered tissue structures, the output voltage of the non-woven fabric under the porcine tissue was about 2.5 times lower than in ethanol, consistent with the decrease in acoustic pressure (Supplementary Figs. 12a and 15b). Also, reflection and absorption at the interface (skin and fat) decreased US acoustic pressure, as further reflected by the decrease in current and voltage (Fig. 2d, Supplementary Fig. 15c). Nevertheless, at 10 cm under layered tissues including skin, fat and muscle, the non-woven fabric yielded a voltage output of >250 mV (right, Fig. 2d). On the other hand, considering the strict application requirements of different neural tissues in the body, we therefore studied the piezoelectricity of ferroelectrets with different sizes. We found a positive correlation between the size of ferroelectrets and their output performance in the range of 1–50 $mm^2$ (Supplementary Fig. 15d, e). Notably, even when the area is as small as 1 $mm^2$, the voltage yielded by the ferroelectret was higher than 100 mV, still capable of effectively stimulating neural tissue[36,37]. These data suggest that the non-woven fabric-based NeuroRing can modulate nervous tissue at multiple depths in vivo.

Considering the scenarios where the NeuroRing would not be encapsulated but directly implanted, we immersed fibrous ferroelectret non-woven fabric in tissue fluid represented by animal tissue fluid without loading electrodes (Fig. 2e). The non-woven fabric exhibited distinct voltage spikes up to ~3 V, which was much stronger than that of the composite fibers as well as the air-blowing fibers (Fig. 2f, g). This demonstrated that the electrical pulses yielded by the ferroelectret fibers were sufficient to stimulate and activate neural tissue.

To verify this, we further applied US to SH-SY5Y-derived neuron-like cells cultured on fibrous ferroelectret non-woven fabric for non-invasive and non-contact stimulation. It is well known that electro-stimulation is able to locally change the membrane potential and trigger the opening of voltage-gated calcium channels, allowing an influx of extracellular $Ca^{2+}$ that activates calmodulin kinases (Fig. 2h)[3,38]. Therefore, we further applied US to SH-SY5Y-derived neuron-like cells cultured on non-woven fabric for non-invasive and non-contact stimulation. We observed that the non-woven fabric was in close contact with the cell, which in this case meant that the released charges can be immediately conducted to cell membranes (Fig. 2i). The $Ca^{2+}$-dependent dye Fluo-4 AM was used to fluorescently stain the cells, in which the fluorescence intensity reflects the intracellular $Ca^{2+}$ concentration to a certain extent (Fig. 2j). Clearly, cells on fibrous ferroelectret non-woven fabric showed significantly enhanced $Ca^{2+}$ expression (Fig. 2k, Supplementary Fig. 16). Together, US-driven electric output from the ferroelectret fibers could facilitate neural stimulation by opening voltage-gated $Ca^{2+}$ channels.

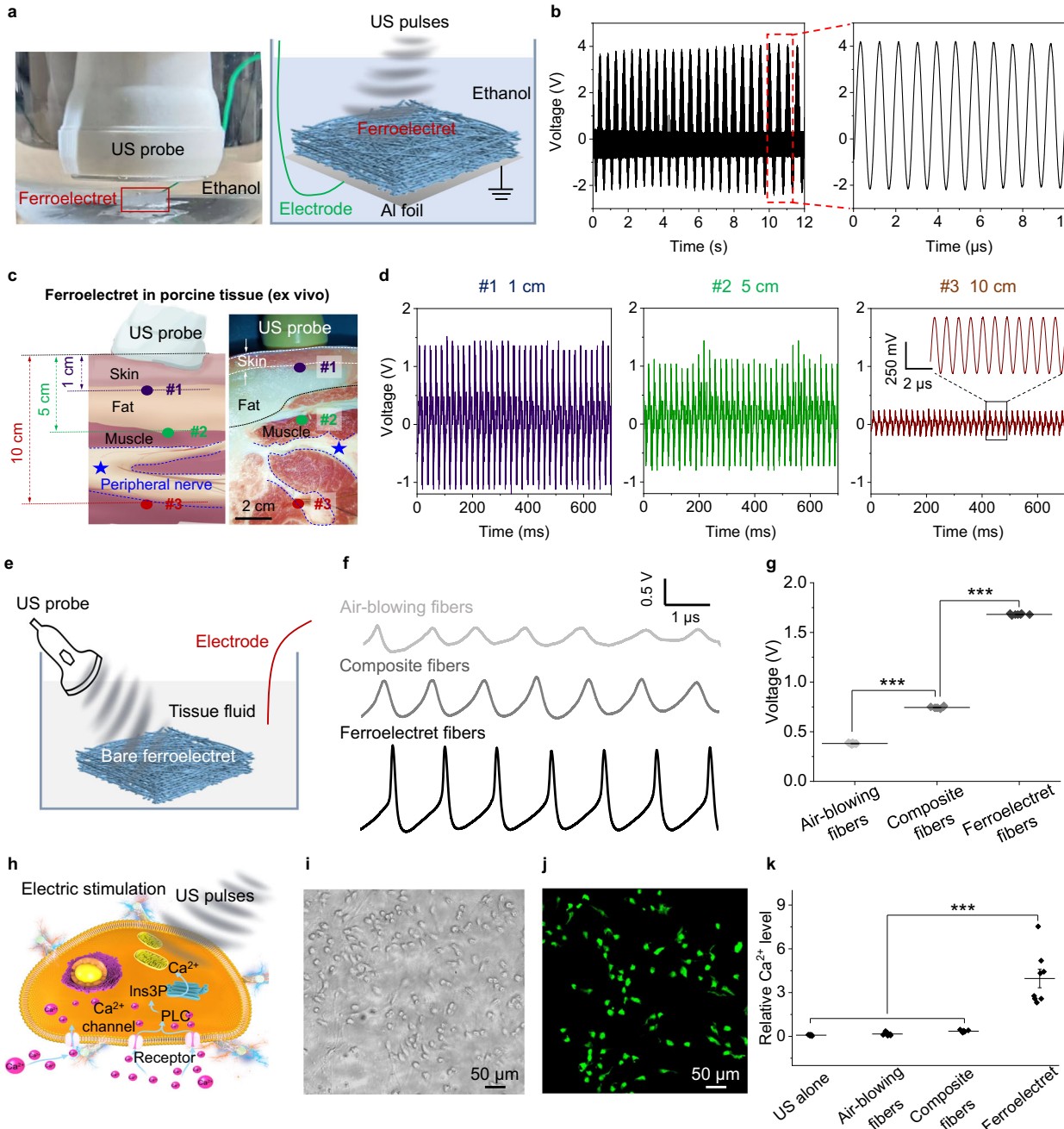

**Fig. 2 | Neuromodulation validation in vitro. a** Experimental setup (right) and schematic diagram (left) for measuring the US-driven electrical output of the ferroelectret in ethanol. **b** Voltage output measured in ethanol at a distance of 10 mm from an US probe to the ferroelectret, with a US setup of 1 MHz and 0.5 W cm⁻². **c** Schematics illustrating ex vivo experiments of ferroelectret implanted in a porcine tissue and visualizing the location of an implanted ferroelectret in porcine tissue. **d** Voltage generated by the ferroelectrets implanted at 1 cm, 5 cm, and 10 cm under the porcine tissue. **e** A schematic of the electrophysiological setup for voltage recording. Voltage change of animal tissue fluid was measured when US pulses were applied to the suspended ferroelectret in animal tissue fluid. **f, g** Voltage outputs yielded from air-blowing fibers, composite fibers, and ferroelectret fibers, respectively. $n = 7$ independent samples for each group. Data are expressed as mean values ± SD. All error bars indicate ± SD. ***$P < 0.001$. Air-blowing fibers/Composite fibers: $P = 6.4439$E-25; Composite fibers/Ferroelectret fibers:

$P = 2.5102$E-32. $P$ values are evaluated through one-sided ANOVA and post-Tukey analysis. **h** Ferroelectret fibers activated by US pulses yield local potentials that alter the membrane potential and/or the configuration of membrane receptors and results in opening of the Ca²⁺ channels. Ins3P, inositol trisphosphate. PLC, phospholipase C. **i** Light microscope image showing cells tightly adhered to the fibers. $n = 8$ independent experiments showing similar results. **j** The fluorescence images of the cells preincubated with Fluo-4 AM (membrane-permeable and Ca²⁺-dependent dye) on the ferroelectret fibers. Green, Ca²⁺. $n = 8$ independent experiments showing similar results. **k** Relative Ca²⁺ levels of cells on air-blowing fibers, composite fibers, and ferroelectret fibers, respectively. Data are expressed as mean values ± SD. All error bars indicate ± SD. $n = 8$ independent samples for each group. ***$P < 0.001$. US alone/Ferroelectret fibers: $P = 2.1786$E-9; Air-blowing fibers/Ferroelectret fibers: $P = 3.4096$E-9; Composite fibers/Ferroelectret fibers: $P = 9.3635$E-9. $P$ values are evaluated through one-sided ANOVA and post-Tukey analysis.

To evaluate its biocompatibility and mechanical match with nerve tissue during in vivo operation, we interfaced the NeuroRing with the sciatic nerve by wrapping fibrous ferroelectric non-woven fabric around the nerve tissue (Fig. 3a). 42 days after implantation, the NeuroRing remained in its original position and tightly wrapped around the nerve, with no signs of degradation in functionality or performance (Fig. 3b, c). Scanning electron microscopy (SEM) was used to further visualize the seamless nerve–NeuroRing interface, confirming their mechanical conformability (Fig. 3d). This stable and conformable interface ensured that undesired detachment or defection occurred during the daily activity of the animal. This would not affect the normal movement and development of rats (Supplementary Fig. 17). Compared to rats implanted with cuff electrodes (the most commonly used electrodes in clinical practice)[39] exhibiting a typical symptom of serious sciatic nerve dysfunction[40], rats implanted with NeuroRing showed normal gait, proper positioning of claws, and toe extension (Fig. 3e, Supplementary Fig. 18). Clearly, the mismatch in mechanical modulus between rigid cuff electrodes and soft nerve tissue deteriorates nerve function, while NeuroRing, which shows a similar modulus to soft neural tissue, completely avoids this flaw. This was further reflected by reconstructed three-dimensional microcomputed tomography ($\mu$CT). The severe restraint of the growth and development of the sciatic nerve by the cuff electrode resulted in structural deformation, while NeuroRing conformedly adhered to the sciatic nerve (Fig. 3f).

Next, nerve tissue was analyzed pathologically using hematoxylin and eosin (H&E) staining. The results revealed that the cuff electrodes exerted severe compression on the nerve tissue, accompanied by the side effects such as epineurial damage, increased vascular density, and decreased nerve bundle size, not seen with the NeuroRing (Supplementary Fig. 19). Also, there was no statistically significant difference in expression levels between NeuroRing-implanted nerves and sham controls of the NF-200 (axon-specific protein), Tuj-1 (a marker for neurofilaments), and tumor necrosis factor-$\alpha$ (TNF-$\alpha$, a marker for inflammatory response) biomarkers (Fig. 3g–i, Supplementary Fig. 20). In contrast, compared to NeuroRing, the expression levels of NF-200 and Tuj-1 were significantly reduced and the expression level of TNF-$\alpha$ was significantly increased in cuff-implanted rats. These results demonstrated that our NeuroRing did not induce damage or an inflammatory response in peripheral neural tissues even under recurrent motion, suggesting excellent biocompatibility to tissue with dynamic motion. Together, structure, gait, and histology results all support the ability of the NeuroRing to accommodate neurodevelopment and locomotion whilst maintaining function.

Pathological tests were also conducted on vital organs, including heart, liver, spleen, lung, kidney, and brain. H&E staining was collected from these organs at different time points (days 1 and 42) after implantation. All the organs showed no deformation or abnormal lymphatic cell invasion (Supplementary Fig. 21), which further confirmed that all rats were in good health condition, and that the NeuroRing had no systemic side effects.

To verify effective neuromodulation in vivo, we carried out stimulation of the sciatic nerve using the NeuroRing triggered by US pulses (Fig. 3j). Concurrently, we recorded real-time electromyographic (EMG) signals from the gastrocnemius muscle innervated by the sciatic nerve. When US was applied, gastrocnemius was activated as reflected in enhanced EMG amplitude (upper curve, Fig. 3k) as well as the power spectral intensity (bottom, Fig. 3k), whereas this was not evident in the US alone group (Supplementary Fig. 22). The level of muscle activation could be quantified by the voltage peak intensity of EMG, with US activation almost two times normal (Fig. 3l). Besides that, muscle activation was also visualized by macroscopic flexion of the ankle joint of $5 \pm 1°$ (Supplementary Fig. 23, Supplementary Movie 2). These results demonstrate that the NeuroRing effectively stimulates the sciatic nerve and modulates its behavior, suggesting promising clinical potential in therapeutic neuromodulation.

Armed with this evidence of peripheral neuromodulation, we went on to explore the efficacy of the NeuroRing in disease treatment. We proposed sacral nerve stimulation to treat acute colitis (top, Fig. 4a). Electrostimulation of sacral nerves is expected to reduce inflammation and pain in colitis due to its direct innervation to the distal colon and rectum (Supplementary Fig. 24a)[41]. This can avoid potential side effects on the cardiovascular system and other internal organs that often occurred with cranial nerves and central nervous system stimulation. Besides, the inhibitory effect of sacral nerve stimulation on colonic inflammation is mediated via the spinal afferent and vagal efferent pathway in addition to the pelvic splanchnic nerve[42,43]. 10 days before induction of colitis, rats underwent NeuroRing implantation that tightly wrapped the splanchnic nerve extending from the S3 sacral nerve (bottom two images in Fig. 4a, Supplementary Fig. 24b). Subsequently, acute colitis was induced by feeding rats with dextran sulfate sodium (DSS) in their drinking water for 7 days. The occurrence of acute colitis can be observed directly from H&E staining (Supplementary Fig. 24c). In this process, we simultaneously performed intervention therapy triggered by US pulses.

To show the involvement of the central nervous system in both spinal afferent and vagal efferent pathways, we measured electroencephalogram (EEG) in the nucleus tractus solitarius (NTS) of the brainstem (Fig. 4b, Supplementary Fig. 25). NTS, a large and complex structure, is the principal site of termination of visceral afferent fibers in the brain. Once ultrasound activates the NeuroRing at the sacral nerve, the EEG signals change and intensify (Fig. 4b, Supplementary Movie 3). To fully evaluate the effect of US-induced electric stimulation, we used normal EEG (rats not receiving such stimulation) and US alone-induced EEG (rats not implanted with NeuroRing) as control groups. Among them, normal EEG as a basic control group was used to observe the influence of US-induced electrical stimulation on brain activity. US alone group was used to evaluate and differentiate the effects of ultrasonic waves and electrical stimulation on brain activity. NeuroRing treatment induced greater activation of neurons in the NTS, compared with US treatment alone (lower curve, Fig. 4b, Supplementary Fig. 26). Besides, thanks to the ring-shaped design, our NeuroRing was not affected by rotational misalignment. Despite some deflection of the ultrasonic transmitter, the nerves can still be effectively regulated, which is further reflected in the EEG signals (Supplementary Fig. 27).

We also verified that the NeuroRing markedly altered the release of sympathetic (norepinephrine (NE)) and vagal endocrine hormones (pancreatic polypeptide (PP)). As shown in Fig. 4c, the NeuroRing significantly enhanced NE and PP secretion in plasma, which didn't occur with US alone. Again, this verified that NeuroRing stimulation of the sacral nerve also activated specific pathways of spinal cord afferent and vagus nerve efferents, synergistically enhancing the anti-inflammatory effect.

Next, we evaluated the therapeutic effect of the NeuroRing on colitis by measuring pro- and anti-inflammatory cytokines in colon tissue (Fig. 4d). It is well known that macrophages play a key role in the progression of DSS-induced colitis[44]. M1 macrophages contribute to DSS-induced colitis by activating pro-inflammatory cytokines, while M2 macrophages activate anti-inflammatory cytokines and maintain tissue remodeling and intestinal homeostasis. CD11b and CD11c cells are considered M1 macrophages, and M2 macrophages can be quantified by F4/80 and CD206. The NeuroRing was found to inhibit M1 macrophage polarization (Fig. 4e, Supplementary Fig. 28a) and promote M2 macrophage polarization (Fig. 4f, Supplementary Fig. 28b). This also explains the decreased expression of inflammation-associated caspase1 in colonic tissue. In contrast, the antioxidant enzyme heme oxygenase-1 (HO-1), which attenuates colitis, was elevated (Fig. 4g, Supplementary Fig. 28c). As a result, the pro-

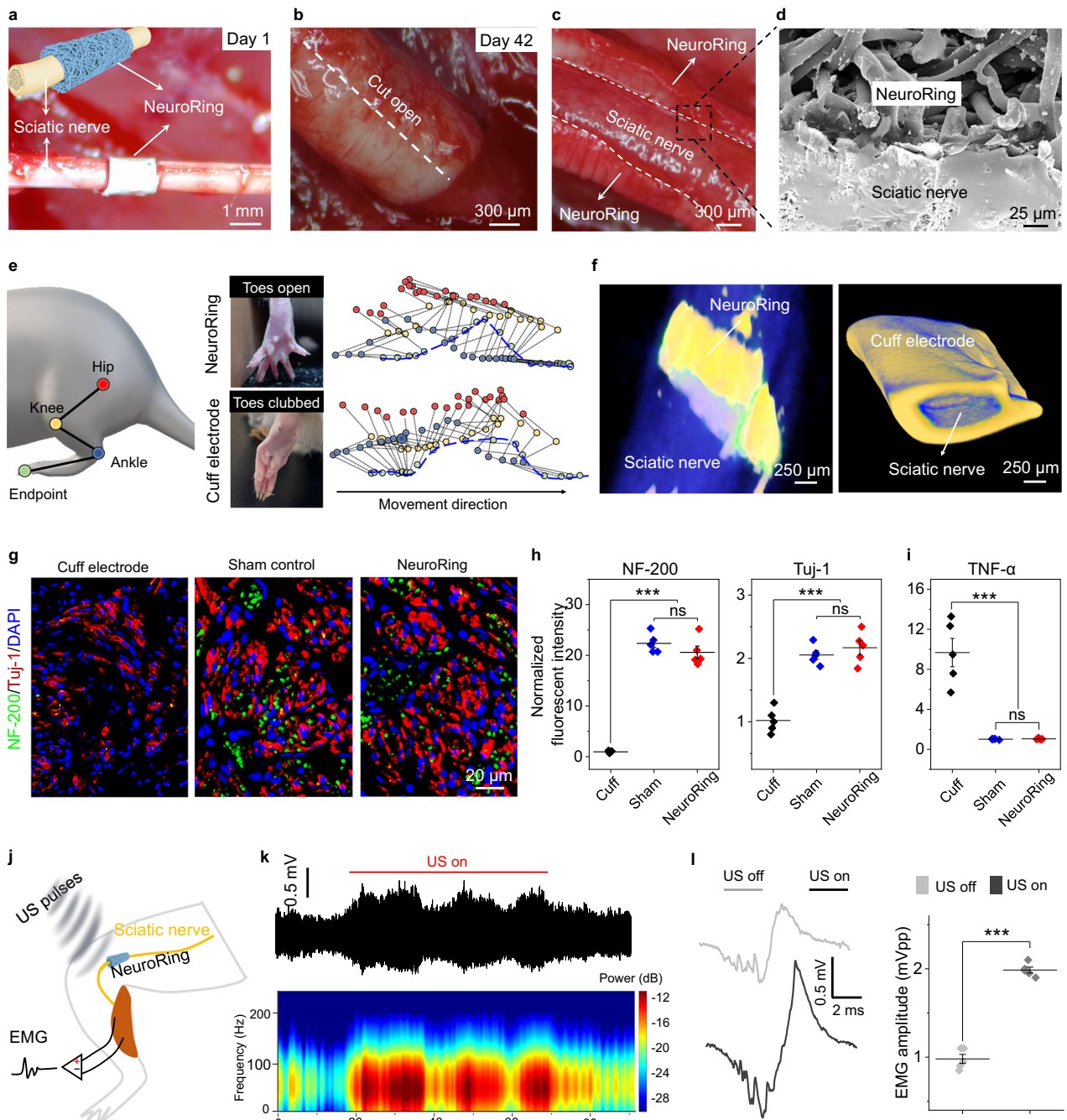

**Fig. 3 | Biocompatibility and behavior of the NeuroRing on sciatic nerves.**
**a** Microscopic image of the implanted NeuroRing wrapped around the sciatic nerve. Inset: Schematic of implantation of the NeuroRing around the sciatic nerve. **b**, **c** Microscopic images of the NeuroRing after 42-day implantation showing it tightly wrapped around the nerve tissue. **d** SEM image showing an intimate NeuroRing–nerve interface. **e** Hindlimb kinematics during walking (left). Images of toes for rats implanted with a NeuroRing and cuff electrode (middle). Stick diagram decompositions of hindlimb movements (right). **f** Reconstructed three-dimensional μCT scans of the sciatic nerve after chronic implantation of NeuroRing and cuff electrode. **g** Triple immunofluorescent staining of NF-200 (green) and Tuj-1 (red), nuclei (blue) of sciatic nerve for the sham control and NeuroRing. **h**, **i** Normalized fluorescence intensity of (**h**) NF-200 (left) and Tuj-1 (right) and (**i**) TNF-α for cuff electrodes, sham control, and NeuroRing. Data are

expressed as mean values ± SD. All error bars indicate ± SD. $n = 5$ biologically independent animals for each group. ***$P < 0.001$, and ns means not significant. NF-200: $P_{Cuff/Sham} = 7.0955E-10$; $P_{Cuff/Neuro} = 1.9283E-9$; $P_{Sham/Neuro} = 0.175$. Tuj-1: $P_{Cuff/Sham} = 3E-6$; $P_{Cuff/Neuro} = 1E-6$; $P_{Sham/Neuro} = 0.392$. TNF-α: $P_{Cuff/Sham} = 8E-6$; $P_{Cuff/Neuro} = 8E-6$; $P_{Sham/Neuro} = 0.979$. $P$ values are evaluated through one-sided ANOVA and post-Tukey analysis. **j** Schematic of EMG performed on gastrocnemius muscle when NeuroRing is triggered by US pulses to stimulate the sciatic nerve. **k** EMG and their corresponding power spectral analysis. **l** Representative EMG signals for US on and US off and their corresponding EMG voltage amplitude. Data are expressed as mean values ± SD. All error bars indicate ± SD. $n = 5$ biologically independent animals for each group. ***$P < 0.001$. $P = 5.7534E-10$. $P$ values are evaluated through one-sided ANOVA and post-Tukey analysis.

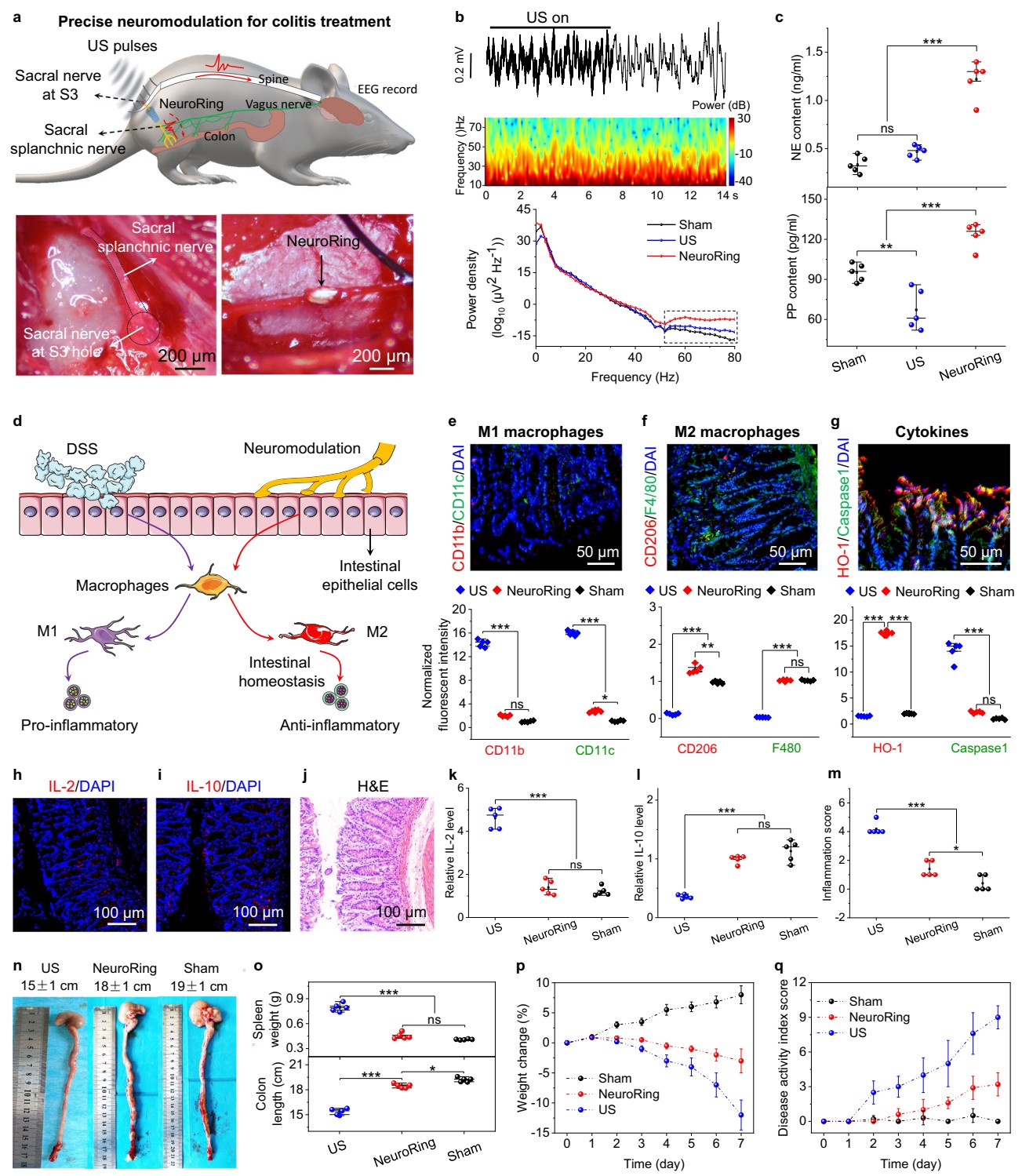

inflammatory cytokine interleukin-2 (IL-2) was suppressed (Fig. 4h, k, Supplementary Fig. 29a) while the anti-inflammatory cytokine IL-10 was potentiated (Fig. 4i, l, Supplementary Fig. 29b) in colonic tissue after NeuroRing treatment. To visualize the effect in colitis treatment, we performed H&E staining of the colon. DSS resulted in extensive colonic tissue injury, shown by inflammatory cell infiltration, crypt damage, and focal formation as compared with the sham control group. These results could be reversed by NeuroRing (Fig. 4j, m, Supplementary Fig. 29c). Also, the expression levels with the NeuroRing were almost equivalent to the Sham group, and significantly better than that in the US alone group, again indicating a significant

therapeutic effect of NeuroRing on colitis. The attenuated severity of colitis was further demonstrated by a longer colon length, lighter spleen weight, less weight loss, and slower increase in disease activity index (DAI) score (Fig. 4n–p). Together, these data demonstrate that the US-triggered NeuroRing can efficiently stimulate sacral splanchnic nerves and precisely modulate colonic tissue, ultimately increasing the resistance of rats to DSS-induced colitis.

## Discussion

In summary, the wireless, leadless, and battery-free NeuroRing is an innovation in biotechnology that has the potential to allow precise

**Fig. 4 | Sacral splanchnic nerve modulation for colitis treatment. a** Schematic and microscopic image of the NeuroRing implant that wrapped around the sacral splanchnic nerve. EEG signals were recorded while ultrasound-induced electrical stimulation of the sacral nerves was performed. **b** EEG (top) recorded over parietal cortex and comparison of the power spectrum (middle) and its derived quantified power spectral density (bottom) for sham control, US alone, and NeuroRing showing the significant enhancement in rats treated with ultrasound-induced electrical stimulation. Sham control refers to the sham operation and normal drinking water group. US alone refers to the sham operation and drinking DSS group. Bottom: the black dashed box intuitively reflects the significant enhancement of ultrasound-induced electrical stimulation, especially in the range of 50–80 Hz. **c** Expression levels of NE (sympathetic endocrine hormones) and PP (vagal endocrine hormones) in blood samples. **d** Schematic representation of the protective mechanism of US-triggered NeuroRing in DSS-induced colitis. Neuromodulation alleviates DSS-induced colitis by inhibiting macrophage M1 polarization and promoting M2 polarization. **e** Immunostaining for M1 macrophages represented by CD11b (red) and CD11c (green) of colon tissue. Bottom: the quantification (bottom) of CD11b and CD11c. CD11b: $P_{US/Neuro} = 1.5801E-15$; $P_{US/Sham} = 6.2921E-16$; $P_{Neuro/Sham} = 0.052$. CD11c: $P_{US/Neuro} = 7.7803E-18$; $P_{US/Sham} = 1.9047E-18$; $P_{Neuro/Sham} = 0.049$. **f** Immunostaining for M2 macrophages represented by CD206 (red) and F4/80 (green). Bottom: the quantification

(bottom) of CD206 and F4/80. CD206: $P_{US/Neuro} = 3.2E-12$; $P_{US/Sham} = 1.6497E-10$; $P_{Neuro/Sham} = 0.009536$. F4/80: $P_{US/Neuro} = 4.4252E-18$; $P_{US/Sham} = 4.4554E-18$; $P_{Neuro/Sham} = 0.962116$. **g** Immunostaining for HO-1 (red) and Caspase1 (green). Bottom: quantification (bottom) of HO-1 and Caspase1. HO-1: $P_{US/Neuro} = 9.3436E-20$; $P_{Neuro/Sham} = 1.3455E-19$. Caspase1: $P_{US/Neuro} = 5.8124E-10$; $P_{US/Sham} = 1.7891E-10$; $P_{Neuro/Sham} = 0.083595$. **h–j** IL-2 (**h**), IL-10 (**i**) immunofluorescent stain, and H&E stain (**j**) images of colon tissue. **k–m** Relative expression level measured from (**k**) IL-2 and (**l**) IL-10 immunofluorescent staining and (**m**) H&E stain images. IL-2: $P_{US/Neuro} = 7.3613E-9$; $P_{US/Sham} = 3.8765E-9$; $P_{Neuro/Sham} = 0.427984$. IL-10: $P_{US/Neuro} = 1E-6$; $P_{US/Sham} = 1.5141E-7$; $P_{Neuro/Sham} = 0.079821$. H&E: $P_{US/Neuro} = 2E-6$; $P_{US/Sham} = 6.8169E-8$; $P_{Neuro/Sham} = 0.019865$. **n** Representative colon pictures for US alone, NeuroRing, and sham control. **o** Average splenic weight and colon length calculated in the different groups. Splenic weight: $P_{US/Neuro} = 2.8504E-9$; $P_{US/Sham} = 7.6726E-10$; $P_{Neuro/Sham} = 0.087571$. Colon length: $P_{US/Neuro} = 1.8128E-9$; $P_{Neuro/Sham} = 0.012544$. **p, q** Average body weight (**p**) and DAI scores (**q**) over time calculated from different groups. Unless otherwise stated, all samples were collected on day 7 after the US intervention. All data are expressed as mean values ± SD. All error bars indicate ± SD. $n = 5$ biologically independent animals for each group. *$P < 0.05$, **$P < 0.01$, ***$P < 0.001$; ns, not significant. All $P$ values are evaluated through one-sided ANOVA and post-Tukey analysis.

modulation of organ function by stimulation of peripheral nerves via ultrasonic pulse triggering. The NeuroRing exhibits high mechanical conformability with nerve tissue without affecting normal development and movement. Our results illustrate that the NeuroRing can precisely modulate the sacral splanchnic nerve to alleviate DSS-induced colitis. This study is a practical demonstration of this electrostimulator. We believe that it has promising clinical potential for targeted peripheral neuromodulation, thereby facilitating precise disease treatment.

This precision technology relies on the visceral nerve atlas that focuses on the innervation of visceral organs[2,7,45], such as the lung, heart, liver, pancreas, kidney, bladder, gastrointestinal tract, and lymphoid and reproductive organs. Their specific innervation, including sympathetic, parasympathetic, sensory, and enteric systems, are still being mapped with the goal of achieving high resolution at the level of nerve fibers and electric impulses. As this visceral neural atlas is gradually refined, it creates exciting potential for precise implantation of the NeuroRing in the clinic. We envision using this atlas as a reference to implant the NeuroRing anywhere in the body for therapeutic effects. The parameters of neurostimulation can be tuned by adjusting US parameters such as frequency and intensity, to meet the multidimensional requirements for different processes evoked via neurological disorders and diseases. Our study of sacral nerve stimulation for colitis is an example of this potential and the solid foundation for electromodulation of peripheral nerves to treat disease, offering a potential alternative to non-specific pharmacological approaches and their attendant side effects[46]. Although our current research is to treat diseases by modulating nerve bundles through fibrous ferroelectric non-woven fabric, the core ideas and supporting technology platforms can be transferred and adapted to disease-specific neurons. For example, the non-woven fabric can be replaced by gel-state nanoparticles that are able to be injected into targeted neural tissue or even where individual neurons are located. This seductive vision is challenging but, conceptually, the neuromodulation protocols introduced here can help bring a new class of precision medicine to patients and serve as a framework for the growing field of bioelectronic medicines.

## Methods
### Preparation of fibrous ferroelectret non-woven fabric
The fibrous ferroelectret non-woven fabric was prepared using a laminar-flow-assisted electrospinning method. The electrospun solution precursor was prepared by dissolving PVDF (MW = 5.34 kDa,

Macklin Biochemical Co., Ltd.) in N, N-Dimethylformamide (DMF, Sinopharm Group Chemical Reagent Co., Ltd.) solution at a concentration of 20% (w/v) under sonication for 90 mins. The ZnO particles with a diameter of 300 ± 50 nm (Macklin Biochemical Co., Ltd.) were then added to PVDF solution to prepare the target electrospun solution such that the ZnO-PVDF ratio was 1:80 (by weight). To obtain a uniform solution, the mixture was processed with a cell disruptor for 60 mins. After defoaming, the resulting solution was used directly for laminar-flow-assisted electrospinning to obtain the as-spun PVDF/ZnO fibers. This process was carried out in a suitable environment (room temperature and relative humidity 35%). The acquired solution was fed into plastic syringes fixed in a microsyringe pump, and the spinning solution was injected through a 22G-nozzle spinneret at a speed of 0.8 ml h$^{-1}$. The electrified liquid droplet was stretched and elongated to fibers from the spinneret via a direct-current, constant-high-voltage (18 kV) power and the coaxially high-speed airflow (airflow velocity of 150 mm s$^{-1}$). The as-spun fibers were collected on an aluminum-coated roller collector at 15 cm from the nozzle. As-received fibers were stored in a vacuum for further experimental operation.

### Material characterization
The morphology and structural distribution were investigated by SEM (Supra 55, Carl Zeiss) and TEM (JEM-2100, JEOL). Elemental composition mapping was performed using an energy-dispersive X-ray detector coupled to the SEM. Before SEM testing, all samples were precoated with gold prior by turbomolecular pumped coater to improve the conductivity (Q150T). The orientation of the polymer chains was visualized by thermogravimetric analysis (TG/SDTA851E, METTLER TOLEDO) and XRD (D8 Advance, Bruker). Stress relaxation and stress-strain curves of ferroelectret film were obtained by using a computer servo tensile tester (HD-B609B-S). The acoustic pressure curve was measured by a hydrophone (Shenzhen Well Come Technology Co.LTD, China).

### Electrical characterization
The voltage signals were measured and recorded using an oscilloscope (UTD2102CAL) with a voltage probe (P5100A, Tektronix, Inc. Beaverton, USA) with 40 MΩ input impedance. US generation and characterization were carried out with a commercial US transducer and generator (WED-101, Well.D Medical Electronics Co., Ltd. Shenzhen, China). The US probe frame was grounded to minimize the noise. The US generator was used to generate US pulses in absolute ethanol and porcine skin.

## Calcium influx during US stimulation

The experiments were divided into three groups, namely air-blowing fibers, composite fibers, and ferroelectret fibers. All samples were soaked in 75% alcohol for 30 mins and dried in a vacuum, then sterilized by ultraviolet light irradiation for 1 h and transferred to the 6-well culture plates. SH-SY5Y cells were seeded onto the samples at a density of $5 \times 10^3$ cells/ml in each sample with DMEM culture medium (10% FBS, 1% penicillin, 1% glutamine, and 0.01% fungizone). Then, the plates were incubated in a 5% $CO_2$ incubator at 37 °C for 3 h to allow cell adhesion. Subsequently, 500 μL fresh medium and Fluo-4 AM (1 μM) were added into each well for further incubation at 37 °C for 30 min. US stimulation was carried out at 0.5 W cm$^{-2}$ for 10 mins after allowing 20 mins stabilization. The intracellular $Ca^{2+}$ dynamics of SH-SY5Y cells were monitored with $Ca^{2+}$ influx imaging during application of US. The Fluo-4 AM fluorescence was monitored by fluorescence microscopy and analyzed with NIKON NIS-Elements advanced research microscope imaging software (Nikon Instruments Inc. Melville, NY, USA).

## Implantation of the NeuroRing to rat sciatic nerves to assess biocompatibility

**NeuroRing implantation and stimulation.** Sprague–Dawley (SD) rats (150–200 g) were fasted for 6 h prior to surgery, then anesthetized with inhalation of 0.8–1.5% isoflurane and maintained with 1.0% isoflurane. Before implantation, the NeuroRing was sterilized by ultraviolet light irradiation for 1 h. After shaving and skin disinfection, the sciatic nerve was exposed by blunt dissection of the vastus lateralis and biceps femoris muscles. The NeuroRing was then wrapped around the sciatic nerve and secured using biologic glue. Muscle and skin were closed with 4-0 and 3-0 nylon sutures, respectively. The NeuroRing was then activated to electrically stimulate the sciatic nerve using an US generator to electrically stimulate the sciatic nerve. Needle electrodes were used to perform EMG on the gastrocnemius muscle, and data were collected using a multi-channel physiological signal acquisition and recording instrument (RM6240, China).

**Immunostaining.** Biocompatibility studies were performed to compare nerve bundle damage and immune responses at the implantation site for both the NeuroRing and sham control. 42 days postoperatively, the sciatic nerve samples wrapped by NeuroRing were taken out. The nerve segments were fixed with 1% tetraoxide, dehydrated, and embedded in Epon812 (American Electron Microscope Science) resin. A cross-section was cut to a thickness of 4 mm (Leica EM UC 6 ultrathin microtome) and mounted on gelatin-precoated slides. The slices were washed with PBS three times and then permeabilized by 0.1% Triton-X 100 in PBS for 15 mins. After washing with PBS solution, the samples were incubated in blocking solution (3% bovine serum albumin, 0.1% Triton-X 100 in PBS) for 45 mins. The samples were then co-stained with 1:400 anti-Tuj-1 (Abcam, USA) and 1:300 anti-NF-200 (Abcam, USA) in blocking solution overnight at 4 °C. The inflammatory response from the NeuroRing was assessed using TNF-α immunofluorescent staining. All slides were evaluated by using an immunofluorescence microscope (Leica, USA).

## Implantation of the NeuroRing to rat splanchnic nerve extending from the S3 sacral nerve to treat colitis

**NeuroRing implantation and stimulation.** SD rats (150–200 g) were fasted for 6 h prior to surgery, then anesthetized with inhalation of 0.8–1.5% isoflurane and maintained with 1.0% isoflurane. Before implantation, the NeuroRing was sterilized by ultraviolet light irradiation for 1 h. A skin incision was made at the coccyx, and the muscles divided to expose the sacral nerve and freeing it from surrounding tissue. The NeuroRing was implanted by wrapping it around the splanchnic nerve extending from the S3 sacral nerve and secured using biological glue. Due to the seamless connection between NeuroRing and nerve tissue, the electrical pulses generated by ultrasound are directly transmitted to the target nerve through the mediation of bodily fluids and tissues. Muscle and skin were then closed with 4-0 and 3-0 nylon sutures, respectively. 10 days after NeuroRing implantation, colitis was induced by the addition of 3.0% wt./vol. DSS to the drinking water for 7 days (from day 0), whereas control rats received normal drinking water. The occurrence of colitis was verified by H&E staining. Synchronously, the NeuroRing was activated to electrically stimulate the sciatic nerve using the US generator to electrically stimulate the sciatic nerve during the feeding of DSS. The US treatment time was 30 minutes in total every day, 15 mins at 10:00 and 15 mins at 20:00, respectively.

**EEG collection.** We acquired EEG signals from NTS of the brainstem when the NeuroRing was activated by US pulses. The EEG was recorded using multi-channel physiological signal acquisition and recording instrument (RM6240). SD rats (150 to 200 g) were fasted for 6 h prior to surgery, then anesthetized with inhalation of 0.8–1.5% isoflurane and maintained with 1.0% isoflurane. The dorsal surface of the skull, including occipital bone and down to the atlanto-atlas juncture, was exposed. For monopolar recording of EEG, stainless-steel screws (1/16″ in diameter) were placed over the dura through small holes drilled over the left frontal (1.5 mm lateral and 0.5 mm rostral to bregma) cortex and secured with dental cement. The electrodes were then connected to stainless-steel screws to transmit and record the rat's EEG signals.

**Measurements of norepinephrine and pancreatic polypeptide.** Blood samples were drawn on day 7 after US treatment. Plasma pancreatic PP (Abcam, USA) and NE (Abcam, USA) were measured by enzyme-linked immunosorbent assay (eBioscience, USA).

**Disease activity index score evaluation.** The animal was observed once daily to calculate a DAI score based on weight loss, stool consistency, and the presence of gross blood in feces. Each item was rated from 0 to 4, yielding a total score from 0 (healthy) to 12 (maximal severity of colitis). The specific criteria for DAI are presented in Supplementary Table 2.

**Colon length and spleen weight.** The colon and spleen were harvested immediately after termination to measure colon length and spleen weight, two important indices of inflammatory response.

**Histology and immunohistology staining.** After 7 days of US treatment, the rats were euthanized and the colons were extracted. Freshly collected samples were washed with PBS 1×, longitudinally cut, positioned as a Swiss Roll in 10% NBF, and incubated at room temperature for 24 h. For H&E staining, all samples were fixed with 4% paraformaldehyde, then embedded in paraffin, and cut into sections with a thickness of 3 μm by using a microtome. For fluorescent immunostaining, all samples were also cut at 3 μm of thickness using a Microtome, left on adhesive slides at room temperature overnight, and baked for 1 h at 65 °C. Antigen retrieval was achieved after 40 min of incubation at 100 °C in antigen unmasking solution (citrate-based, pH = 6.0; Vector Laboratories, Burlingame, CA), followed by 15 mins incubation with SuperBlock Blocking Buffer (Thermo Scientific, Waltham, MA) at room temperature. After blocking, samples were incubated overnight at 4 °C. The slices were incubated with the primary antibody: 1:200 anti-CD11b (Abcam, USA) and 1:200 anti-CD11c (Abcam, USA), 1:200 anti-CD206 (Abcam, USA) and 1:200 anti-F/480 (Abcam, USA), 1:200 anti-HO-1 (Abcam, USA) and 1:200 anti-Caspase1 (Abcam, USA), 1:200 anti-IL6 (Abcam, USA) and 1:200 anti-IL-10 (Abcam, USA). Then, the secondary antibody corresponding to the species of the primary antibody was added to cover the tissues at room temperature, protected from light for 4 h. Finally, all images were acquired on an immunofluorescence microscope (Leica, USA). The grading of histological damage was evaluated according to Supplementary Table 3.

All rats received humane care and were handled in accordance with the Institutional Animal Care and Use Committee approval protocol of the Animal Care Center at the Nanjing Jinling Hospital.

## Reporting summary
Further information on research design is available in the Nature Portfolio Reporting Summary linked to this article.

## Data availability
All data supporting the findings of this study are available within the article and its supplementary files. Any additional requests for information can be directed to, and will be fulfilled by, the corresponding author(s). Source data are provided with this paper.

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

## Acknowledgements

Z.-Q.F. acknowledges financial support from the National Natural Science Foundation of China (82302406, 51773093, and 11204033), F.J. and T.L. acknowledge financial support from China Postdoctoral Science Foundation (2023M731696, 2022TQ0158, and 2022M721616), F.J. and T.L. acknowledge financial support from Jiangsu Funding Program for Excellent Postdoctoral Talent (2023ZB539 and 2022ZB250), and Z.-Q.F. and F.J. acknowledge financial support from the Fundamental Research Funds for the Central Universities (30923010307 and 30920041105).

## Author contributions

T.L. and Z.-Q.F. conceived the idea and designed the research. T.L. and Z.D.W. performed most experiments. T.L., Z.D.W., L.S.H. and J.M. performed the animal experiments. T.L., F.J., H.B.W., W.Y.Z., L.L.Q. and H.H.Z. performed electrical signal acquisition and analysis. H.D.G. and T.L. performed theoretical simulations and analyses. T.L., Y.Y., T.W., J.M., M.I., S.W, Z.K.W. and Z.-Q.F. analyzed the data and wrote the manuscript. All authors contributed to discussions on the data and commented on the manuscript.

## Competing interests

The authors declare no competing interests.
