## [Peer Review File · Nature Communications]

REVIEWER COMMENTS

Reviewer #1 (Remarks to the Author):

In this article, the authors have reported a wireless, leadless, and battery-free ferroelectret implant, termed NeuroRing, that wraps around the target peripheral nerve and demonstrates high mechanical conformability to dynamic motion nerve tissue. As-fabricated NeuroRing can act as an ultrasound receiver that converts ultrasound vibrations into electrostimulation pulses, thus stimulating the targeted peripheral nerve on demand. This work is interesting, but some questions should be addressed.

- 1、 The authors just told the reader the US with a frequency of 1 MHz and an intensity of 0.5 W cm⁻². What is the ultrasound strength applied in the experiment? Please measure the ultrasound strength applied on the soft ferroelectret, which is very important for neuro modulation.
- 2、 Line 119-121, the authors mentioned that "Remarkably, even at 10 cm under layered tissues including skin, fat and muscle, the non-woven fabric yielded a voltage output of > 250 mV (right, Fig. 2d), still capable of effectively stimulating neural tissue (> 72 mV)." Please give the reference to prove it.
- 3、 For Supplementary Video 2, it is cannot prove that muscle activation is caused by ultrasound-induced electrical stimulation. Hand pressure also can make the ankle joint move. This animal experiment may be done keeping the leg hanging in the air,
- 4、 For EEG signal, to the best of our knowledge, it can be measured at any time. How to prove the effect of ultrasound-induced electric stimulation? The comparative experiment should be carried out.
- 5、 Ultrasound can directly open the Ca²⁺ channels. How to prove the opening of the Ca²⁺ channels is just due to ultrasound-induced electrical stimulation?
- 6、 For the colitis treatment, please tell more details about how to place the NeuroRing and connect the electrode and nerve. The Nerve is usually stimulated by DC electrical signal. In this experiment, no rectifier is utilized. How does it work? The recorded EEG signal is very weak and it is difficult to prove the effect of ultrasound-induced electric stimulation. Please explain it.

Reviewer #2 (Remarks to the Author):

- Since this is a passive implantable device, the accurate control of its stimulation parameters (e.g., strength) is very challenging considering alignment issues and tissue loss. It is also harder to scale it up to distributed implants for multi-site stimulation.

- The main novelty of this work is on developing new soft ultrasonic materials. Its application for nerve stimulation is less significant considering the available devices. Therefore, the authors must do a better

job in providing a comprehensive comparison of the performance of this material with the relevant state-of-the-art materials.

- Is there any study for the stability and biocompatibility of these materials in long-term operation (well more than couple of weeks)?

- Measurement result in Fig. 2: The external transducer was driven at 1 MHz? Why was this frequency chosen? Is the implanted flexible piezoelectric device resonating? Its dimension is 5 mm which is much lower than 1 MHz.

- Measurement result in Fig. 2: The authors claim that these voltages are enough for neural stimulation. But these are 1 MHz pulses. For successful stimulation, kHz pulses are often applied to a nerve. Also, the current injected to the nerve is important (not the voltage necessary). What is the injected current (or alternatively the electric field) applied to the nerve in these conditions?

- Measurement result in Fig. 2b: At what frequency did you pulse the 1 MHz ultrasound transducer? What is the source of large background voltage when the pulse is zero?

- Since the ultrasound sonication (thereby electrical pulses) was at 1 MHz, what was the underlying mechanism for neuromodulation in in vivo tests/results in Fig. 3 and 4. Also, what is the estimated voltage on the implant in these tests?

Reviewer #3 (Remarks to the Author):

The first observation that drives the decision on this article relates to novelty of the work done. The foundation of the paper revolves around the use of PVDF/ZnO film obtained through electrospinning as the ferroelectret. However, it is well-known that electrospinning can prepare PVDF nanofibers (e.g., *Sensors (Basel)*. 2020 Sep; 20(18): 5214). Incorporating nanoparticles like ZnO is not uncommon in such studies. Additionally, this reviewer has doubts about the claim that their electrospun PVDF/ZnO can be considered a ferroelectret device. Firstly, their voids appear to be very small (Fig. 1d), which, according to calculations and previous experiments by ferro electret nanogenerators (FENG) researchers, may not generate sufficient piezoelectric effects.

Secondly, information about the poling process using high voltage was not explained clearly. Even if the authors performed poling, there is no comparison provided between samples with and without high-voltage poling. This reviewer questions whether the enhanced piezoelectric effect is solely due to the intrinsic piezoelectricity of PVDF after the addition of nanoparticles and/or stretching, which has been known to enhance the piezoelectric properties.

The second main observation relates to the discrepancy between the results obtained in this work and prior art. The size of this ferroelectret (5 mm by 5mm) is very unlikely to produce the 6 V mentioned in

this article, especially if the target tissue is under human skin. Although it has been found that the response from ferroelectrets increases with frequencies, and get good response in the ultrasonic regions (in published articles that the current manuscript ignores), it is very difficult to understand how such a small device under human skin is capable of responding to ultrasonic frequencies with such large voltage response. Along this same line, where are the current plots? Did the authors try to characterize output short circuit current from these devices? Did the authors consider the matching impedance between the device and the instrument used to measure the generated voltage? Obtaining real electro-mechanical measurements from ferroelectrets is not a trivial task, and this reviewer believes that these experiments/studies will generate questions about the electrical voltage output that they are measuring –which could be due to sources other than the ferroelectret.

Finally, although their application may appear fancy, it takes a quick look to the references and relevant prior art to realize that this is also not a novel concept –this reviewer has personally read this same application roughly 5 years ago. Lastly, it's disheartening to note that they not only failed to cite any recent relevant work (including a very recent comprehensive review article on ferro electrets and their applications), but also neglected to reference important studies by other well-known researchers in the field of ferroelectrets.

Reviewer #1

In this article, the authors have reported a wireless, leadless, and battery-free ferroelectret implant, termed NeuroRing, that wraps around the target peripheral nerve and demonstrates high mechanical conformability to dynamic motion nerve tissue. As-fabricated NeuroRing can act as an ultrasound receiver that converts ultrasound vibrations into electrostimulation pulses, thus stimulating the targeted peripheral nerve on demand. This work is interesting, but some questions should be addressed.

Response: We appreciate the reviewer's high recognition and professional comments.

1. The authors just told the reader the US with a frequency of 1 MHz and an intensity of 0.5 W cm^{-2} . What is the ultrasound strength applied in the experiment? Please measure the ultrasound strength applied on the soft ferroelectret, which is very important for neuromodulation.

Response: Good point. Per the reviewer's request, we used a hydrophone (Shenzhen Well Come Technology Co.LTD, China) to measure the ultrasound strength (i.e., acoustic pressure) applied to the ferroelectret. Firstly, we measured the acoustic pressure on the ferroelectret film in ethanol corresponding to **Fig. 2a** in the manuscript. The acoustic pressure applied to the ferroelectret device (dimension of 5 mm by 5 mm) is 117 kPa (**Revised Supplementary Fig. 12a**). Subsequently, the acoustic pressure on the ferroelectret inside the pork tissue was measured corresponding to **Fig. 2c** in the manuscript. As the implantation depth increases, the acoustic pressure shows the same trend as the voltage (**Revised Supplementary Fig. 15b**). Considering that this work is applied to the sacral nerve about 1 cm under the skin of rats, the acoustic pressure applied on the NeuroRing around the nerve is about 30 kPa.

Revised Supplementary Fig. 12. a, Experimental setup (left) and acoustic pressure curve (right) measured in ethanol at a distance of 10 mm from an US probe to the acoustic pressure probe, with a US setup of 1 MHz and 0.5 W cm^{-2} .

Revised Supplementary Fig. 15. b, Acoustic pressure applied on the ferroelectrets implanted at 1 cm, 5 cm, and 10 cm under the porcine tissue.

2. Line 119-121, the authors mentioned that “Remarkably, even at 10 cm under layered tissues including skin, fat and muscle, the non-woven fabric yielded a voltage output of > 250 mV (right, Fig. 2d), still capable of effectively stimulating neural tissue (> 72 mV).” Please give the reference to prove it.

Response: Many thanks for the reviewer’s reminder. For greater rigor, **references 35 and 36** have been added to support our statement in the **revised manuscript**.

New added references in revised manuscript:

36. Lewis R, Asplin KE, Bruce G, Dart C, Mobasheri A, Barrett-Jolley R. The role of the membrane potential in chondrocyte volume regulation. *J Cell Physiol* **226**, 2979-2986 (2011).
37. Zaszczynska A, Sajkiewicz P, Gradys A. Piezoelectric Scaffolds as Smart Materials for Neural Tissue Engineering. *Polymers (Basel)* **12**, (2020).

3. For Supplementary Video 2, it cannot prove that muscle activation is caused by ultrasound-induced electrical stimulation. Hand pressure also can make the ankle joint move. This animal experiment may be done keeping the leg hanging in the air.

Response: We are grateful for the reviewer’s rigor and professionalism. To avoid ambiguity, we used a frame to fix the ultrasound probe and suspend it in the air, without manual operation (**Revised Supplementary Fig. 23a**). Moreover, there is an ultrasound coupling agent between the ultrasound probe and the rat skin, which also excludes the impact of the probe's own weight (**Revised Supplementary Fig. 23b**). Subsequently, as requested by the reviewer, we kept the rat's leg hanging in the air before applying ultrasound (**Revised Supplementary Fig. 23a**). Finally, after applying ultrasound, we still found macroscopic flexion of the ankle joint of $5 \pm 1^\circ$ (**Revised Supplementary Fig. 23c**). Synchronously, we also have updated the **revised Supplementary Video 2** with the rat's leg dangling in the air.

Revised Supplementary Fig. 23. **a**, Snapshot photo of a suspended and fixed ultrasound probe. **b**, Snapshot photo of the ultrasound coupling agent between the ultrasound probe and the rat skin. **c**, Representative images of the movement of the legs before and after stimulation. Average angular change of the ankle joint in response to stimulation.

4. For EEG signal, to the best of our knowledge, it can be measured at any time. How to prove the effect of ultrasound-induced electric stimulation? The comparative experiment should be carried out.

Response: Another good point. To fully evaluate the effect of ultrasound-induced electric stimulation, we used normal EEG (rats not receiving such stimulation) and US alone-induced EEG (rats not implanted with NeuroRing) as control groups. By comparing the results between these groups, we can assess the specific effects and benefits of ultrasound-induced electric stimulation. Clearly, the intensity (frequency) of EEG signals were significantly enhanced in rats treated with ultrasound-induced electrical stimulation, but not significantly changed in rats treated with US alone without NeuroRing implantation (**top two images in Fig. 4b, Revised Supplementary Fig. 26**). This was more intuitively reflected from the quantified power spectral density derived from the EEG power spectrum, especially from 50 to 80 Hz, as marked by the black dashed box in the bottom image of **revised Fig. 4b. (page 10, line 2 to 7 in revised manuscript)**

Also, in the original manuscript, from the data of neurohormone secretion and physiological markers as well as colitis treatment, compared with the control groups including US alone and sham groups, the US-induced electric stimulation shows significant advantages (**Fig. 4c, Fig. 4e to q, Supplementary Figs. 28 and 29**). This further indirectly confirms the effect of ultrasound-induced electric stimulation.

Revised Supplementary Fig. 26. EEG (top) recorded over parietal cortex and comparison of power spectrum (bottom) for US alone. US alone refers to the sham operation and drinking DSS group.

Revised Fig. 4. b, EEG (top) recorded over parietal cortex and comparison of power spectrum (middle) and its derived quantified power spectral density (bottom) for sham control, US alone and NeuroRing showing the significant enhancement in rats treated with ultrasound-induced electrical stimulation. Sham control refers to the sham operation and normal drinking water group. US alone refers to the sham operation and drinking DSS group. Bottom: the black dashed box intuitively reflects the significant enhancement of ultrasound-induced electrical stimulation, especially in the range of 50 to 80Hz.

5. Ultrasound can directly open the Ca^{2+} channels. How to prove the opening of the Ca^{2+} channels is just due to ultrasound-induced electric stimulation?

Response: In our study, we simultaneously cultured cells on air-blowing PVDF fibers, PVDF/ZnO composite fibers, and ferroelectret fibers. It is worth noting that the determination of intracellular Ca^{2+} on different materials was carried out during the application of ultrasonic excitation. Clearly, cells on ferroelectret fibers showed significantly enhanced Ca^{2+} expression compared to the other two materials (**Fig. 2k**). To further verify the effect of ultrasound-induced electric stimulation yielded from ferroelectret fibers, we added a new control group of ultrasound excitation alone without any materials. Although ultrasonic stimulation alone also caused an influx of calcium ions, it was almost negligible compared with piezoelectric materials including blown PVDF fibers, PVDF/ZnO composite fibers, and ferroelectric electret fibers (**Revised Fig. 2k, Revised Supplementary Fig. 16**). Therefore, from these data, we can conclude that although sound waves are able to open Ca^{2+} channels,¹⁻³ in our study, ultrasound-induced electric stimulation is the main factor for the opening of Ca^{2+} channels.

Revised Fig. 2. k, Relative Ca^{2+} levels of cells on air-blowing fibers, composite fibers and ferroelectret fibers under ultrasound stimulation. All error bars indicate \pm SD. $n = 8$ for Ca^{2+} expression level statistics. $***P < 0.001$.

Revised Supplementary Fig. 16. Relative Ca^{2+} levels of cells on air-blowing fibers and composite fibers under ultrasound stimulation.

References:

1. Przystupski D, Ussowicz M. Landscape of Cellular Bioeffects Triggered by Ultrasound-Induced Sonoporation. *Int J Mol Sci* **23**, (2022).
 2. Qin P, et al. Sonoporation-induced depolarization of plasma membrane potential: analysis of heterogeneous impact. *Ultrasound Med Biol* **40**, 979-989 (2014).
 3. Takahashi T, Nakagawa K, Tada S, Tsukamoto A. Low-energy shock waves evoke intracellular Ca(2+) increases independently of sonoporation. *Sci Rep* **9**, 3218 (2019).
6. For the colitis treatment, please tell more details about how to place the NeuroRing and connect the electrode and nerve. The Nerve is usually stimulated by DC electrical signal. In this experiment, no rectifier is utilized. How does it work? The record EEG signal is very weak and it is difficult to prove the effect of ultrasound-induced electric stimulation. Please explain it.

Response: We thank the reviewer's comment. It is necessary to point out that the NeuroRing, which works in a wireless and electrodeless mode, is implanted by directly wrapping it around the splanchnic nerve extending from the S3 sacral nerve as shown in **Fig. 4a**. This is because the implant site can directly innervate the distal colon and rectum, reducing inflammation and pain in colitis.^{1,2} Due to the seamless connection between NeuroRing and nerve tissue, the electrical pulses generated by ultrasound are directly transmitted to the target nerve through the mediation of bodily fluids and tissues. To this end, we focused on describing the implantation details and working principles in the 'Methods' section as follows: '*SD rats (150 to 200 g) were fasted for 6 h prior to surgery, then anesthetized with inhalation of 0.8–1.5% isoflurane and maintained with 1.0% isoflurane. Before implantation, the NeuroRing was sterilized by ultraviolet light irradiation for 1 h. A skin incision was made at the coccyx, and the muscles divided to expose the sacral nerve and freeing it from surrounding tissue. The NeuroRing was implanted by wrapping it around the splanchnic nerve extending from the S3 sacral nerve and secured using biologic glue. Due to the seamless connection between NeuroRing and nerve tissue, the electrical pulses generated by ultrasound are directly transmitted to the target nerve through the mediation of bodily fluids and tissues. Muscle and skin were then closed with 4-0 and 3-0 nylon sutures, respectively.*' (page 15, line 12-21 in revised manuscript)

For the second comment, actually, the basic waveforms used in electrotherapy, including nerve stimulation, are generally divided into three groups: direct current (DC), pulsed DC, and alternating current (AC). Notably, in comparison with DC, AC electrotherapy is safer and better subjectively tolerated by the patient, and its DC component is always zero, which prevents chemical damage of the skin/tissue/organ.⁷ AC electrotherapy allows also for long-lasting applications in vivo. Therefore, many implantable electrotherapy devices such as pacemakers, cochlear implants, and essentially all other chronically implanted neuroelectronic devices used in clinical settings rely on charge-balanced, biphasic pulses or other forms of AC to excite neural or muscular activity.⁷⁻¹¹ In our study, under the excitation of ultrasound, NeuroRing can directly convert the sound waves into AC signals to stimulate nerve tissue without rectifiers (**Fig. 2a-g**). Immediately afterwards, these AC signals will act on nerve cells, inducing them to yield action potentials, and ultimately stimulate and modulate neural tissue (**Fig. R1**).

For the third comment, we recommend viewing **revised Supplementary Video 3**. to get a quick impression of the extraordinary features of our study. Once ultrasound activates the NeuroRing at the sacral nerve, the EEG signals change and intensify. This is further quantified by the quantified power spectral density derived from the EEG signal (**lower curve, revised Fig. 4b**). In the range

of 50 to 80 Hz, the EEG of ultrasound-induced electric stimulation shows the greatest power density, which means the most active brain activity. This is consistent with the conclusion that NeuroRing shows significant advantages in treating colitis.

Fig. R1. NeuroRing can be activated by ultrasound vibrations to yield electric impulses. On the basis of the resting potential, the electric impulses will stimulate the nerve cells in contact with NeuroRing, and trigger the local cells to produce a transmissible membrane potential fluctuation (i.e., action potential). After the action potential is generated, it is not limited to the local site stimulated, but propagates rapidly along the plasma membrane until the cells in the whole tissue produce an action potential in turn. Action potentials in nerve tissue play a central role in cell–cell communication. They propagate signals along the neuron's axon toward synaptic boutons which then connect with other neurons at synapses, or to motor cells or glands. Virtually all organs and their functions are regulated through such nerve impulses.

Revised Fig. 4. b, EEG (top) recorded over parietal cortex and comparison of power spectrum (middle) and its derived quantified power spectral density (bottom) for sham control, US alone and NeuroRing showing the significant enhancement in rats treated with ultrasound-induced electrical stimulation. Sham control refers to the sham operation and normal drinking water group. US alone refers to the sham operation and drinking DSS group. Bottom: the black dashed box intuitively reflects the significant enhancement of ultrasound-induced electrical stimulation, especially in the range of 50 to 80Hz.

References:

1. Tu L, Gharibani P, Zhang N, Yin J, Chen JD. Anti-inflammatory effects of sacral nerve stimulation: a novel spinal afferent and vagal efferent pathway. *Am J Physiol Gastrointest Liver Physiol* **318**, G624-G634 (2020).
2. Furness JB. The enteric nervous system and neurogastroenterology. *Nat Rev Gastroenterol Hepatol* **9**, 286-294 (2012).
3. Golanov EV, Reis DJ. Neurons of nucleus of the solitary tract synchronize the EEG and elevate cerebral blood flow via a novel medullary area. *Brain Res* **892**, 1-12 (2001).
4. Chase MH, Nakamura Y, Clemente CD, Sterman MB. Afferent vagal stimulation: neurographic correlates of induced EEG synchronization and desynchronization. *Brain Res* **5**, 236-249 (1967).
5. Zabara J. Inhibition of experimental seizures in canines by repetitive vagal stimulation. *Epilepsia* **33**, 1005-1012 (1992).
6. Grill HJ, Hayes MR. The nucleus tractus solitarius: a portal for visceral afferent signal processing, energy status assessment and integration of their combined effects on food intake. *Int J Obes (Lond)* **33 Suppl 1**, S11-15 (2009).
7. Fridman GY, Della Santina CC. Safe direct current stimulator 2: concept and design. *Annu Int Conf IEEE Eng Med Biol Soc* **2013**, 3126-3129 (2013).
8. Guenther T, Lovell NH, Suaning GJ. Bionic vision: system architectures: a review. *Expert Rev Med Devices* **9**, 33-48 (2012).
9. Wilson BS, Dorman MF. Cochlear implants: a remarkable past and a brilliant future. *Hear Res* **242**, 3-21 (2008).
10. Jin F, et al. Biofeedback electrostimulation for bionic and long-lasting neural modulation. *Nat Commun* **13**, 5302 (2022).
11. Jin F, et al. Physiologically Self-Regulated, Fully Implantable, Battery-Free System for Peripheral Nerve Restoration. *Adv Mater* **33**, e2104175 (2021).

Reviewer #2

1. Since this is a passive implantable device, the accurate control of its stimulation parameters (e.g., strength) is very challenging considering alignment issues and tissue loss. It is also harder to scale it up to distributed implants for multi-site stimulation.

Response: We highly appreciate the reviewer's professional comments and sharp scientific insights. Almost all passive strategies for implantable devices, typically such as inductive power transfer, radio frequency energy transfer and acoustic (ultrasound) power transfer, suffer from alignment problems and tissue loss.¹⁻⁴ For ultrasound receivers, there are still several main challenges: 1) interfacial losses due to acoustic impedance mismatches, 2) the reliance of receivers on rigid inorganic piezoelectric materials, and 3) the harsh efficiency penalty for transducer-receiver misalignment. Therefore, the required physical contact between devices and tissue, as well as losses due to impedance mismatches and misalignment, must always be considered during system design. These challenges can be addressed by novel functional materials with enhanced piezoelectric and acoustic properties.

By decoupling the above three issues, we designed and fabricated a ring-shaped soft fibrous ferroelectret (i.e., NeuroRing). It facilitates precise control of stimulation parameters (e.g., intensity) and modulation of nerves. Details are as follows:

- 1) **Fibrous non-woven fabric designed for reducing acoustic impedance.** A mismatch in acoustic impedance is common between transducer materials and the tissue because these materials have different average acoustic velocity and densities. When two materials have large differences between acoustic impedance ultrasound signals are reflected at the interface, leading to wave reflection and reduced powering efficiency.⁵ For example, water has an acoustic impedance of 1.5 MRayl, and human soft tissue demonstrates an acoustic impedance of 1.63 MRayl.⁶ The typical values ceramics are >30 MRayl for bulk or plate-type piezoelectric and <3 MRayl for polymeric piezoelectric.^{7,8} Clearly, soft polymeric piezoelectric materials can effectively reduce acoustic impedance. On the other hand, many studies have found that fiber polymers have lower acoustic impedance and can even resolve acoustic impedance mismatches.^{9,10} This has been widely verified in nature, for example, in the human auditory system, the tympanic membrane is responsible for resolving the acoustic impedance mismatch between the air of the ear canal and the fluid of the inner ear.¹¹ Therefore, we used electrospun fibers-based non-woven fabrics to construct the NeuroRing.
- 2) **High-performance ferroelectret with tissue-like mechanical modulus.** Existing ultrasonic transducers almost all rely on rigid inorganic piezoelectric ceramics, mainly lead-based, owing to their high piezoelectric properties.¹²⁻¹⁵ However, they are mechanically mismatched to soft neural tissue, which causes insertion-related lesions, inflammation reactions, and even neuronal apoptosis, ultimately even leading to therapeutic failure.^{16,17} Although much effort has attempted to overcome these limitations by developing piezoelectric composite materials that combine piezoelectric ceramics and polymers, they have been unsuccessful as either piezoelectricity or flexibility and processability are compromised. Polymeric piezoelectric materials, represented by poly(vinylidene fluoride) (PVDF), offer superior flexibility and excellent biocompatibility, but their weak piezoelectricity largely limits their electric physiotherapy as effective receivers, unless flexibility is sacrificed.¹⁸⁻²⁰ To address these issues, we introduce a porous ferroelectret structure by utilizing the cavitation effect between

inorganic particles and soft electrospun PVDF matrix, which can obtain additional electric dipoles at the surface of the material and inside it, thus enhancing the electrical performance output while maintaining flexibility. The piezoelectric coefficient ($-d_{33}$) we measured is as high as $56 \pm 2 \text{ pC N}^{-1}$, considerably higher than that of piezoelectric polymers as well as even comparable to piezoelectric ceramics.²¹⁻²³ Meanwhile, Young's modulus value is estimated to be 7.6 MPa which is in the same order as soft tissue.²⁴ This modulus value is four to six orders of magnitude lower than that of commonly used piezoelectric ceramics such as PZT and BaTiO₃, and its piezoelectric coefficient was 1 to 2 orders of magnitude higher than that of piezoelectric polymers with similar modulus value (**Fig. 1f, Revised Supplementary Table 1**). These two points ensure that our NeuroRing attaches perfectly to soft nerve tissue and converts ultrasound pulses into electrical pulses efficiently.

- 3) **Ring-shaped design for improving alignment.** A disadvantage of this approach is that ultrasonic waves propagate directionally. As a result, slight misalignments and misorientations between the external transducer and the implanted receiver lead to reductions in coupling efficiency.²⁵ The positioning and form (e.g., pulse frequency) of the ultrasound transmitter are critical for effective power delivery, which can be adjusted based on the location and positioning of the ultrasound receiver.²⁶ However, for implanted ultrasound receivers, especially small ones (millimeter size), regulating external ultrasound transmitters is obviously difficult and complex. Fortunately, research has found that designing novel receiver geometries, such as cylinders of piezoelectric materials, holds promise in increasing tolerance to rotational misalignment efforts.^{27,28} Considering that the application scenario of our study is peripheral nerve modulation, the high-performance fibrous ferroelectret film is designed in a ring shape that is able to tightly wrap around the nerve. We measured the ultrasound intensity applied to the ring-shaped devices and found that they were not affected by rotational misalignment. When excited by 0.5 W cm^{-2} ultrasound pulses with a frequency of 1 MHz, normal incident ultrasound produces a similar ultrasound intensity ($0.2 \pm 0.01 \text{ W cm}^{-2}$) irrespective of ultrasound transmitter rotation (**black curve in revised Supplementary Fig. 13b**). At the same time, we simultaneously recorded the voltage outputs and found that they were also unaffected by rotational misalignment as long as the distance between the ultrasound probe and the fiber membrane was the same (**blue curve in revised Supplementary Fig. 13b**). This is crucial for precise peripheral neuromodulation. (**page 5, line 9 to 12 in revised manuscript**)

Thanks to these optimizations, our fibrous ferroelectret non-woven fabric were able to efficiently convert ultrasound vibrations into electric impulses. It can even yield a voltage output of more than 250 mV at a depth of 10 cm under layered tissues including skin, fat and muscle, which was sufficient to stimulate peripheral nervous system in humans from a few millimeters to more than ten centimeters (covering most of the peripheral nerves). Besides that, when our NeuroRing is implanted into the body of rats, despite some deflection of the ultrasonic transmitter, the nerves can still be effectively regulated, which is further reflected on the EEG signals (**revised Supplementary Fig. 27**). (**page 10, line 9 to 11 in revised manuscript**)

Indeed, multi-site stimulation is a drawback of our technology, and we appreciate the reviewer's constructive reminder. This promotes a more in-depth and detailed engineering design of our NeuroRing in the application of neurostimulation. The possible feasible solutions we envision are: 1) Reducing the width of the NeuroRing and implanting multiple devices in parallel in the same area; 2) Designing multiple parallel arranged small unit NeuroRings on a single device through

hollowed-out pattern method. In future research, we will further validate these two hypotheses. We hope that through these two designs, our NeuroRing has the potential to expand into multi-site stimulation applications in the human body in the future.

Revised Supplementary Table 1. Comparison of d_{33} piezoelectric coefficient between ceramics, polymers, and state-of-the-art composites.

Classification	Materials	$ d_{33} $ (pC N ⁻¹)	Young's modulus (MPa)	Reference
Ceramics	Aluminum nitride (AlN)	5	204,000-396,000	12,13
	Barium Titanate (BaTiO ₃)	45-191	80,000-200,000	14,15,16
	Lithium niobate (LiNbO ₃)	6-16	140,000-355,000	17,18
	Zinc oxide (ZnO)	9.9-26.7	35,000-140,000	19,20,21,22
	Potassium sodium niobate (KNN)	138.2-180	100,000-180,000	23,24
	KNN- LiSbO ₃	283	64,940	25,26
	Lead Zirconate Titanate (PZT)	90-870	70,000-139,000	27
	Aerosol-deposited PZT	406	60,000	28
	Micro-fabricated PZT	/	3,400	29
	Bismuth sodium titanate (BNT)	12.5-150	100,000	30,31
	Cadmium sulfide (CdS)	2.56, 10.65	70,000	32,33
Polymers	Glycine	5.3	2000-9000	34
	Cellulose	19.3	15,000-20,000	35,36,37
	Parylene-C	1-2	4500	38,39
	β-CN	16.5	2500	40
	Diphenylalanine (DPA)	17.9	19,000-27,000	41
	Poly(Lactic Acid) (PLLA)	7-12	1.06-20,000	42,43,44,45
	PVDF-TrFE	25-40	1,500	46,47
	Nylon-11	7.2	/	48
	Polyimide	2.7	/	49
	PVDF film	13-58.5	400-1800	50,51,52,53,54,55,56,57
	Porous PVDF	/	14.6-121	58

	PVDF fibers	25-28	1.64-91.8	59,60
Composites	KNN-PVDF	12	470-1480	61,62
	PVDF/BaTiO ₃	48	2170-3030	63,64
	PVDF/ZnO	31.4	2170	65
	PVDF/PZT	35	1500-2000	66
	Polypropylene/Silicates	/	1500	67
	PVDF/PDMS	/	0.8-30.16	68
	P(VDF-TrFE)/BaTiO ₃	46	784	69,70
	PVDF/ZnO fibers	56 ± 2	7.6	This work

Revised Supplementary Fig. 13. b, The relationship between ultrasound intensity on ring-shaped ferroelectret film and the deflection angle of ultrasound probe.

Revised Supplementary Fig. 27. After the ultrasonic incidence angle was deflected by 60°, the nerve was still activated, which was reflected in the unchanged EEG signal compared with the vertical excitation.

References:

1. Basaeri H, Christensen DB, Roundy S. A review of acoustic power transfer for bio-medical implants. *Smart Materials and Structures* 25, (2016).
2. Turner BL, et al. Ultrasound-Powered Implants: A Critical Review of Piezoelectric Material Selection and Applications. *Adv Healthc Mater* 10, e2100986 (2021).
3. Singer A, Robinson JT. Wireless Power Delivery Techniques for Miniature Implantable Bioelectronics. *Adv Healthc Mater* 10, e2100664 (2021).
4. Rosa BMG, Anastasova S, Yang G-Z. Feasibility Study on Subcutaneously Implanted Devices in Male Rodents for Cardiovascular Assessment Through Near - Field Communication Interface. *Advanced Intelligent Systems* 3, (2021).
5. Rakita A, Nikolic N, Mildner M, Matiasek J, Elbe-Burger A. Re-epithelialization and immune cell behaviour in an ex vivo human skin model. *Sci Rep* 10, 1 (2020).
6. Altun B, Demirkan I, Isik EO, Kocaturk O, Unlu MB, Garipcan B. Acoustic impedance measurement of tissue mimicking materials by using scanning acoustic microscopy. *Ultrasonics* 110, 106274 (2021).
7. Rathod VT. A Review of Acoustic Impedance Matching Techniques for Piezoelectric Sensors and Transducers. *Sensors (Basel)* 20, (2020).
8. Fahad BM, Al-Jadiri RSF. Acoustic Impedance Evaluation of the Polymer-Polymer Hybrid Composites as Insulator Building Materials. *Polymers (Basel)* 15, (2023).
9. Yan W, et al. Single fibre enables acoustic fabrics via nanometre-scale vibrations. *Nature* 603, 616-623 (2022).
10. Parker NG, Mather ML, Morgan SP, Povey MJ. Longitudinal acoustic properties of poly(lactic acid) and poly(lactic-co-glycolic acid). *Biomed Mater* 5, 055004 (2010).
11. Fay JP, Puria S, Steele CR. The discordant eardrum. *Proc Natl Acad Sci U S A* 103, 19743-19748 (2006).
12. Radziemski L, Makin IR. In vivo demonstration of ultrasound power delivery to charge implanted medical devices via acute and survival porcine studies. *Ultrasonics* 64, 1-9 (2016).
13. Hindrichsen CG, Lou-Møller R, Hansen K, Thomsen EV. Advantages of PZT thick film for MEMS sensors. *Sensors and Actuators A: Physical* 163, 9-14 (2010).
14. Basaeri H, Yu Y, Young D, Roundy S. A MEMS-Scale Ultrasonic Power Receiver for Biomedical Implants. *IEEE Sensors Letters* 3, 1-4 (2019).
15. Dagdeviren C, et al. Conformal piezoelectric energy harvesting and storage from motions of the heart, lung, and diaphragm. *Proc Natl Acad Sci U S A* 111, 1927-1932 (2014).
16. Surmenev RA, et al. Hybrid lead-free polymer-based nanocomposites with improved piezoelectric response for biomedical energy-harvesting applications: A review. *Nano Energy* 62, 475-506 (2019).
17. Wu Y, Ma Y, Zheng H, Ramakrishna S. Piezoelectric materials for flexible and wearable electronics: A review. *Materials & Design* 211, (2021).
18. Li J, et al. Multifunctional Artificial Artery from Direct 3D Printing with Built-In Ferroelectricity and Tissue-Matching Modulus for Real-Time Sensing and Occlusion Monitoring. *Adv Funct Mater* 30, (2020).
19. Yang F, et al. Wafer-scale heterostructured piezoelectric bio-organic thin films. *Science* 373, 337-342 (2021).
20. Liu W, Cheng X, Fu X, Stefanini C, Dario P. Preliminary study on development of PVDF nanofiber based energy harvesting device for an artery microrobot. *Microelectronic Engineering* 88, 2251-2254 (2011).
21. Wang F, Mai Y-W, Wang D, Ding R, Shi W. High quality barium titanate nanofibers for flexible piezoelectric device applications. *Sensors and Actuators A: Physical* 233, 195-201 (2015).

22. Cordero F. Quantitative evaluation of the piezoelectric response of unpoled ferroelectric ceramics from elastic and dielectric measurements: Tetragonal BaTiO₃. *Journal of Applied Physics* 123, (2018).
23. Maruyama K, Kawakami Y, Narita F. Young's modulus and ferroelectric property of BaTiO₃ films formed by aerosol deposition in consideration of residual stress and film thickness. *Japanese Journal of Applied Physics* 61, (2022).
24. Patil AC, Thakor NV. Implantable neurotechnologies: a review of micro- and nanoelectrodes for neural recording. *Med Biol Eng Comput* 54, 23-44 (2016).
25. Won SM, Cai L, Gutruf P, Rogers JA. Wireless and battery-free technologies for neuroengineering. *Nat Biomed Eng*, (2021).
26. Ozeri S, Shmilovitz D. Ultrasonic transcutaneous energy transfer for powering implanted devices. *Ultrasonics* 50, 556-566 (2010).
27. Sun Y, Gao X, Wang H, Chen Z, Yang Z. A wideband ultrasonic energy harvester using 1-3 piezoelectric composites with non-uniform thickness. *Applied Physics Letters* 112, (2018).
28. Barbruni GL, Ros PM, Demarchi D, Carrara S, Ghezzi D. Miniaturised Wireless Power Transfer Systems for Neurostimulation: A Review. *IEEE Trans Biomed Circuits Syst* 14, 1160-1178 (2020).

2. The main novelty of this work is on developing new soft ultrasonic materials. Its application for nerve stimulation is less significant considering the available devices. Therefore, the authors must do a better job in providing a comprehensive comparison of the performance of this material with the relevant state-of-the-art materials.

Response: We thank the reviewer's comment and constructive suggestion. First of all, we agree with the reviewer's point of view that the novelty of this work is the design and preparation of new soft electromechanical coupling materials for ultrasound receivers. To comprehensively and objectively prove the superiority of our materials, we conducted a thorough literature review and comparisons with the relevant state-of-the-art materials, covering ceramics, polymers and their composites (**Fig. 1f**). For details, please find them in **revised Supplementary Table 1**. As we all know, ultrasound receiver can convert ultrasound into electrical signals through piezoelectric elements.¹⁻⁵ It is expected to be used in vivo as an adjunct or alternative to drugs to treat neurological disorders and diseases via stimulating the peripheral nervous system on demand triggered by ultrasound. However, existing ultrasonic transducers almost all rely on rigid inorganic piezoelectric ceramics, mainly lead-based, owing to their high piezoelectric properties.⁶⁻⁹ They are mechanically mismatched to soft neural tissue, which causes insertion-related lesions, inflammation reactions, and even neuronal apoptosis, ultimately even leading to therapeutic failure.^{10,11} Many efforts have attempted to develop piezoelectric composites combining inorganic particles and polymers to exploit their advantages and overcome their limitations. However, their properties, such as piezoelectricity and flexibility, are still far below expectations. The 'rule of mixture' is often used to predict the properties of ideal composites, but it appears to have been broken in the design and development of functional piezoelectric hybrids, with either piezoelectricity or flexibility and processability being compromised. The problem is that even the tight bond between the inorganic particles and the polymer matrix cannot improve the material's properties substantially enough for practical applications while retaining its flexibility. For example, as reported in *Nature* in 2022, Yan et al. developed a ferroelectret composed of a mixture of PVDF-TrFE and BaTiO₃ with a high piezoelectric coefficient up to 46 pC N⁻¹, but at the expense of the flexibility of the polymer matrix.¹² It is still mechanically mismatched to soft neural tissue. While polymeric piezoelectric materials, represented by PVDF, offer superior flexibility and

excellent biocompatibility, their weak piezoelectricity largely limits their applications as effective ultrasound receivers.¹³⁻¹⁵ Innovatively, we introduce a porous ferroelectret structure by utilizing the cavitation effect between inorganic particles and soft electrospun PVDF matrix, which can obtain additional electric dipoles at the surface of the material and inside it, thus enhancing the electrical performance output while maintaining flexibility. The piezoelectric coefficient ($-d_{33}$) we measured is as high as $56 \pm 2 \text{ pC N}^{-1}$, considerably higher than that of piezoelectric polymers as well as even comparable to piezoelectric ceramics.¹⁶⁻¹⁸ Meanwhile, Young's modulus value is estimated to be 7.6 MPa which is in the same order as soft tissue.¹⁹ This modulus value is four to six orders of magnitude lower than that of commonly used piezoelectric ceramics such as PZT and BaTiO₃, and its piezoelectric coefficient was 1 to 2 orders of magnitude higher than that of polymers with similar modulus value. Therefore, taken together, we break down the common barrier wall in piezoelectric mixtures of inorganic particles and polymer matrix—the incompatibility of piezoelectricity and flexibility. Our materials achieve the trade-off between flexibility and piezoelectricity, maintaining excellent flexibility while also achieving extremely high piezoelectricity.

On the other hand, in addition to innovation of materials, it is necessary to point out that our NeuroRing also has significant advantages over currently available neurostimulation devices. Our paper presents a novel approach to peripheral neuromodulation using a soft ring-shaped ultrasound receiver. This innovative device offers several advantages over existing technologies, including high mechanical conformability to soft nerve tissue, wireless operation, and reduced risk of injury and inflammation. The development of tissue-matched ultrasound receivers is not only important for advancing our understanding of neurological disorders but also for improving the lives of millions of people around the world who suffer from these conditions. By supporting this technology, we can help to bring about a new era of targeted treatments for neurological disorders. *For details, please refer to the response to comment 1.*

Revised Supplementary Table 1. Comparison of d_{33} piezoelectric coefficient between ceramics, polymers, and state-of-the-art composites.

Classification	Materials	$ d_{33} $ (pC N ⁻¹)	Young's modulus (MPa)	Reference
Ceramics	Aluminum nitride (AlN)	5	204,000-396,000	12,13
	Barium Titanate (BaTiO ₃)	45-191	80,000-200,000	14,15,16
	Lithium niobate (LiNbO ₃)	6-16	140,000-355,000	17,18
	Zinc oxide (ZnO)	9.9-26.7	35,000-140,000	19,20,21,22
	Potassium sodium niobate (KNN)	138.2-180	100,000-180,000	23,24
	KNN- LiSbO ₃	283	64,940	25,26
	Lead Zirconate Titanate (PZT)	90-870	70,000-139,000	27

	Aerosol-deposited PZT	406	60,000	28
	Micro-fabricated PZT	/	3,400	29
	Bismuth sodium titanate (BNT)	12.5-150	100,000	30,31
	Cadmium sulfide (CdS)	2.56, 10.65	70,000	32,33
Polymers	Glycine	5.3	2000-9000	34
	Cellulose	19.3	15,000-20,000	35,36,37
	Parylene-C	1-2	4500	38,39
	β -CN	16.5	2500	40
	Diphenylalanine (DPA)	17.9	19,000-27,000	41
	Poly(Lactic Acid) (PLLA)	7-12	1.06-20,000	42,43,44,45
	PVDF-TrFE	25-40	1,500	46,47
	Nylon-11	7.2	/	48
	Polyimide	2.7	/	49
	PVDF film	13-58.5	400-1800	50,51,52,53,54,55,56,57
	Porous PVDF	/	14.6-121	58
	PVDF fibers	25-28	1.64-91.8	59,60
Composites	KNN-PVDF	12	470-1480	61,62
	PVDF/BaTiO ₃	48	2170-3030	63,64
	PVDF/ZnO	31.4	2170	65
	PVDF/PZT	35	1500-2000	66
	Polypropylene/Silicates	/	1500	67
	PVDF/PDMS	/	0.8-30.16	68
	P(VDF-TrFE)/BaTiO ₃	46	784	69,70
	PVDF/ZnO fibers	56 ± 2	7.6	This work

References:

1. Won SM, Cai L, Gutruf P, Rogers JA. Wireless and battery-free technologies for neuroengineering. *Nat Biomed Eng*, (2021).
2. Curry EJ, et al. Biodegradable nanofiber-based piezoelectric transducer. *Proc Natl Acad Sci U S A* 117, 214-220 (2020).
3. Liu Y, et al. Exercise-induced piezoelectric stimulation for cartilage regeneration in rabbits. *Sci Transl Med* 14, eabi7282 (2022).
4. Pop F, Herrera B, Rinaldi M. Lithium Niobate Piezoelectric Micromachined Ultrasonic Transducers for high data-rate intrabody communication. *Nat Commun* 13, 1782 (2022).
5. Zhang T, et al. Piezoelectric ultrasound energy-harvesting device for deep brain stimulation and analgesia applications. *Sci Adv* 8, eabk0159 (2022).

6. Radziemski L, Makin IR. In vivo demonstration of ultrasound power delivery to charge implanted medical devices via acute and survival porcine studies. *Ultrasonics* 64, 1-9 (2016).
7. Hindrichsen CG, Lou-Møller R, Hansen K, Thomsen EV. Advantages of PZT thick film for MEMS sensors. *Sensors and Actuators A: Physical* 163, 9-14 (2010).
8. Basaeri H, Yu Y, Young D, Roundy S. A MEMS-Scale Ultrasonic Power Receiver for Biomedical Implants. *IEEE Sensors Letters* 3, 1-4 (2019).
9. Dagdeviren C, et al. Conformal piezoelectric energy harvesting and storage from motions of the heart, lung, and diaphragm. *Proc Natl Acad Sci U S A* 111, 1927-1932 (2014).
10. Surmenev RA, et al. Hybrid lead-free polymer-based nanocomposites with improved piezoelectric response for biomedical energy-harvesting applications: A review. *Nano Energy* 62, 475-506 (2019).
11. Wu Y, Ma Y, Zheng H, Ramakrishna S. Piezoelectric materials for flexible and wearable electronics: A review. *Materials & Design* 211, (2021).
12. Yan W, et al. Single fibre enables acoustic fabrics via nanometre-scale vibrations. *Nature* 603, 616-623 (2022).
13. Li J, et al. Multifunctional Artificial Artery from Direct 3D Printing with Built-In Ferroelectricity and Tissue-Matching Modulus for Real-Time Sensing and Occlusion Monitoring. *Adv Funct Mater* 30, (2020).
14. Yang F, et al. Wafer-scale heterostructured piezoelectric bio-organic thin films. *Science* 373, 337-342 (2021).
15. Liu W, Cheng X, Fu X, Stefanini C, Dario P. Preliminary study on development of PVDF nanofiber based energy harvesting device for an artery microrobot. *Microelectronic Engineering* 88, 2251-2254 (2011).
16. Wang F, Mai Y-W, Wang D, Ding R, Shi W. High quality barium titanate nanofibers for flexible piezoelectric device applications. *Sensors and Actuators A: Physical* 233, 195-201 (2015).
17. Cordero F. Quantitative evaluation of the piezoelectric response of unpoled ferroelectric ceramics from elastic and dielectric measurements: Tetragonal BaTiO₃. *Journal of Applied Physics* 123, (2018).
18. Maruyama K, Kawakami Y, Narita F. Young's modulus and ferroelectric property of BaTiO₃ films formed by aerosol deposition in consideration of residual stress and film thickness. *Japanese Journal of Applied Physics* 61, (2022).
19. Patil AC, Thakor NV. Implantable neurotechnologies: a review of micro- and nanoelectrodes for neural recording. *Med Biol Eng Comput* 54, 23-44 (2016).
20. Won SM, Cai L, Gutruf P, Rogers JA. Wireless and battery-free technologies for neuroengineering. *Nat Biomed Eng*, (2021).
21. Burton A, et al. Wireless, battery-free subdermally implantable photometry systems for chronic recording of neural dynamics. *Proc Natl Acad Sci U S A* 117, 2835-2845 (2020).
22. Wei Z, et al. Physical Cue - Based Strategies on Peripheral Nerve Regeneration. *Advanced Functional Materials*, (2022).
23. Jin F, et al. Biofeedback electrostimulation for bionic and long-lasting neural modulation. *Nat Commun* 13, 5302 (2022).
24. Kavvadias T, Huebner M, Brucker SY, Reisenauer C. Management of device-related complications after sacral neuromodulation for lower urinary tract disorders in women: a single center experience. *Arch Gynecol Obstet* 295, 951-957 (2017).
25. del Valle J, Navarro X. Interfaces with the peripheral nerve for the control of neuroprostheses. *Int Rev Neurobiol* 109, 63-83 (2013).
26. Mickle AD, et al. A wireless closed-loop system for optogenetic peripheral neuromodulation. *Nature* 565, 361-365 (2019).
27. Piech DK, et al. A wireless millimetre-scale implantable neural stimulator with ultrasonically powered bidirectional communication. *Nat Biomed Eng* 4, 207-222 (2020).

28. Chen JC, et al. A wireless millimetric magnetoelectric implant for the endovascular stimulation of peripheral nerves. *Nat Biomed Eng* 6, 706-716 (2022).
29. Zheng H, et al. A shape-memory and spiral light-emitting device for precise multisite stimulation of nerve bundles. *Nat Commun* 10, 2790 (2019).
30. Liu Y, et al. Soft and elastic hydrogel-based microelectronics for localized low-voltage neuromodulation. *Nat Biomed Eng* 3, 58-68 (2019).

3. Is there any study for the stability and biocompatibility of these materials in long-term operation (well more than couple of weeks)?

Response: We thank the reviewer's comment. Per the reviewer's request, we provided stability and biocompatibility data for ferroelectret materials. Actually, prior to being used as a NeuroRing for neuromodulation, our ferroelectret materials underwent stability testing and a 6-month subcutaneous biocompatibility testing. Our ferroelectret materials demonstrate excellent long-term stability and biocompatibility. Specifically,

For the stability study, we placed ferroelectret materials soaked in phosphate-buffered saline in a thermostat gas bath vibrator (CHA-SA, China) at 37 °C for 6 months. During this period, we tested the piezoelectric output of the ferroelectret materials every month. Before testing, these materials were washed three times with absolute ethanol to remove residual buffer, then immersed in absolute ethanol and excited using an ultrasonic generator (1 MHz, 0.5 W cm⁻²). The test parameters are the same as the original manuscript. Simply, the distance between the ultrasonic probe and the ferroelectret film is 10 mm. The test device operated in single-electrode mode using an aluminum foil as the electrode, and a copper wire conducting the charges, as shown in **Fig. 2a**. The results were presented in **revised Supplementary Fig. 13a**. It can be seen that the ferroelectret materials have excellent stability, and the output peak-to-peak voltage is maintained at 6 ± 1 V with almost no change. (**Page 5, line 7 to 9 in revised manuscript**)

As for the biocompatibility study, before implantation in rats, we first performed cytocompatibility testing in vitro. Neural stem cells were cultured on the ferroelectret materials and petri dishes (gold standard). After 3 days of culture, the expression level of Ki67 was assessed (**Fig. R2a**). Ki67 is a related antigen of proliferated cells and can be used as marker of proliferation ability. It can be seen that the cell proliferation on the ferroelectret materials did not exhibit significant differences compared to the culture dishes (**Fig. R2b**). These data provide preliminary confirmation that our ferroelectret materials were biocompatible and nontoxic. Then, to further verify the long-term biocompatibility in vivo, the ferroelectret materials were implanted into the gastrocnemius muscle area and around the sciatic nerve of the rats for nearly 6 months (**Fig. R3a**). This region was associated with greater muscle rhythmicity, which could diagnose the potential infections and necrosis in surrounding tissues and evaluate the mechanical stability of the materials in vivo. The implants were removed at the 1st, 3rd, 6th and 24th weeks after implantation. Histological analyses were performed by staining prepared tissue slides with hematoxylin and eosin (**Fig. R3b**). The histological images of the implantation area showed a very mild immune reaction without significant presence of inflammation and cellular toxicity. Fibrosis and activated macrophages were found in the 1st week, improved from week 3, and reduced to the normal level at week 6, and continued until the 24th week. These data confirmed long-term biocompatibility of our ferroelectret materials.

Revised Supplementary Fig. 13. a, Comparison of the output voltage before and after the ferroelectret fibers being immersed in a PBS solution at 37 °C for 6 months.

Fig. R2. a, Representative images of Ki67 (proliferation marker)-marked cells cultured on ferroelectret materials and petri dish in proliferation medium for 3 days. **b**, Relative Ki67 expression levels of cells on ferroelectret materials and petri dish. $n = 5$ for Ki67 expression level statistics.

Fig. R3. Surgical image showing the implantation of ferroelectret films into the gastrocnemius muscle area of rats for the histological analysis, and hematoxylin and eosin staining of gastrocnemius muscles at the implanted area. Asterisks (*) show locations of the implantation.

4. Measurement result in Fig. 2: The external transducer was driven at 1 MHz? Why was this frequency chosen? Is the implanted flexible piezoelectric device resonating? Its dimension is 5 mm which is much lower than 1 MHz.

Response: We thank the reviewer's professional comment. Yes, the measurement results in **Fig. 2** were taken at an ultrasonic frequency of 1 MHz. We selected 1 MHz after comprehensive consideration of various factors, such as biosafety, spatial resolution and electromechanical coupling efficiency, etc. For now, actually, many studies choose ultrasound with low frequencies < 1 MHz. This is because low frequencies can improve the efficiency of energy transfer by reducing tissue-mediated attenuation,¹⁻³ compensating to a certain extent the low piezoelectric performance of implanted ultrasound receivers. However, low frequencies often cause deleterious tissue effects, such as cavitation and inevitable heating,⁴⁻⁶ which prone to cause apoptosis, tissue collapse, or even necrosis, and thus disrupt the physiological function of normal tissues. Also, low frequencies accompanied by low resolution will result in a large focal region, making it difficult to accurately locate the precise stimulation area.² This is a critical issue, especially for micron-sized peripheral nerves. According to the principle of acoustics, high-frequency ultrasound has the ability to generate small focal region, thereby improving the accuracy of stimulation.⁷ Narrowing the stimulation area with high-frequency ultrasound will provide good opportunities to expand its application.^{8,9} For example, focused ultrasound transducer with 1 MHz frequency will generate a focal width of about 4.3 mm. What cannot be ignored is that at high frequencies, especially above 10 MHz, the absorption of ultrasound by biological tissues becomes substantial, resulting in extremely severe ultrasound attenuation.^{10,11} Therefore, the common ultrasound frequency range used for neuromodulation is 0.25 to 1.0 MHz.^{10,12-16} Under this premise, as long as our ultrasonic receiver has sufficiently high piezoelectric performance, we choose a higher frequency of 1 MHz. In fact, many studies do the same, 1 MHz was chosen as the driving frequency.^{1,17}

As for whether our ultrasonic receiver resonates with ultrasonic waves with a frequency of 1 MHz, we tested and counted the voltage output of ferroelectrets driven by ultrasound at frequencies ranging from 200 kHz to 1400 kHz (**Fig. R4**). It can be inferred that the resonant frequency of our fibrous ferroelectret is around 700 kHz due to the high voltage output at this frequency. Thanks to the high performance of our ferroelectric materials, it is ensured that the ultrasound receiver can generate sufficient electrical stimulation pulses at non-resonant frequencies. Of course, we appreciate the constructive questions from the reviewers, which inspired us to customize electromechanical coupling materials that resonate with high-frequency ultrasound (>1 MPa) in future research to further improve piezoelectric properties and biocompatibility.

Fig. R4. Normalized voltage output of ferroelectrets driven by different ultrasonic frequencies.

References:

1. Zhang T, et al. Piezoelectric ultrasound energy-harvesting device for deep brain stimulation and analgesia applications. *Science Advances* 8, (2022).
2. Li G-F, et al. Improved Anatomical Specificity of Non-invasive Neuro-stimulation by High Frequency (5 MHz) Ultrasound. *Scientific Reports* 6, (2016).
3. Bystritsky, A. et al. A review of low-intensity focused ultrasound pulsation. *Brain Stimul* 4, 125–136 (2011).
4. Turner BL, et al. Ultrasound - Powered Implants: A Critical Review of Piezoelectric Material Selection and Applications. *Advanced Healthcare Materials* 10, (2021).
5. O'Brien, W. D. Jr. Ultrasound-biophysics mechanisms. *Prog. Biophys. Mol. Biol.* 93, 212–255 (2007).
6. Nelson, T. R., Fowlkes, J. B., Abramowicz, J. S. & Church, C. C. Ultrasound biosafety considerations for the practicing sonographer and sonologist. *J. Ultrasound Med.* 28, 139–150 (2009).
7. Foster FS, Pavlin CJ, Harasiewicz KA, Christopher DA, Turnbull DH. Advances in ultrasound biomicroscopy. *Ultrasound in Medicine & Biology* 26, 1-27 (2000).
8. Poon, A. S. Y., O'Driscoll, S. & Meng, T. H. Optimal frequency for wireless power transmission into dispersive tissue. *IEEE Trans. Antennas Propag.* 58, 1739–1750 (2010).
9. Kurs, A. et al. Wireless power transfer via strongly coupled magnetic resonances. *Science* 317, 83–86 (2007).
10. Riis T, Kubanek J. Effective Ultrasonic Stimulation in Human Peripheral Nervous System. *IEEE Transactions on Biomedical Engineering* 69, 15-22 (2022).
11. R. S. Cobbold, *Foundations of Biomedical Ultrasound*. London, U.K.: Oxford Univ. Press, (2006).
12. Naor O, Krupa S, Shoham S. Ultrasonic neuromodulation. *Journal of Neural Engineering* 13, (2016).
13. Fomenko A, Neudorfer C, Dallapiazza RF, Kalia SK, Lozano AM. Low-intensity ultrasound neuromodulation: An overview of mechanisms and emerging human applications. *Brain Stimulation* 11, 1209-1217 (2018).

14. Legon W, et al. Transcranial focused ultrasound modulates the activity of primary somatosensory cortex in humans. *Nature Neuroscience* 17, 322-329 (2014).
15. Lee W, Kim H, Jung Y, Song I-U, Chung YA, Yoo S-S. Image-Guided Transcranial Focused Ultrasound Stimulates Human Primary Somatosensory Cortex. *Scientific Reports* 5, (2015).
16. Lee W, et al. Transcranial focused ultrasound stimulation of human primary visual cortex. *Scientific Reports* 6, (2016).
17. Pop F, Herrera B, Rinaldi M. Lithium Niobate Piezoelectric Micromachined Ultrasonic Transducers for high data-rate intrabody communication. *Nature Communications* 13, (2022).

5. Measurement result in Fig. 2: The authors claim that these voltages are enough for neural stimulation. But these are 1 MHz pulses. For successful stimulation, kHz pulses are often applied to a nerve. Also, the current injected to the nerve is important (not the voltage necessary). What is the injected current (or alternatively the electric field) applied to the nerve in these conditions?

Response: Good point. In fact, many previous studies have successfully utilized MHz electrical pulses (≥ 1 MHz) for neuromodulation such as peripheral nerve stimulation and deep brain stimulation.¹⁻⁵ For example, Taejeong Kim et al. used piezoelectric particles driven by ultrasound at a frequency of 1 MHz for deep brain stimulation in the treatment of Parkinson's disease.¹ Eli J. Curry et al. used 1MHz ultrasound triggered piezoelectric materials for deep brain stimulation to open the blood-brain barrier.² Joshua C. Chen et al. successfully activated the sciatic nerve of rats using a 1.25 MHz magnetoelectric pulse.⁵ Although pulses with frequencies of MHz pulses can be effectively used for neural regulation, the specific mechanism is still not clearly elucidated.^{1,6} We are very grateful for the reviewer's comment, which is also a question we have been thinking about. A common hypothesis is that neuromodulation electrically stimulates the nervous system by opening voltage-gated ion channels (such as Ca^{2+} , Na^+ , and K^+) in nearby nerve cells and further alters pathological network activity.⁷⁻⁹ *Our study is consistent with this hypothesis that ultrasound-induced electrical pulses promoted the opening of Ca^{2+} channels in SH-SY5Y-derived neuron-like cells (Fig. 2h to k).*

We agree with the reviewer that the current injected into the nerve is also important. To this end, we supplemented the current-related data. The test method is the same as the corresponding voltage test method under pork tissue as shown in **Fig. 2c and d**. We used a current probe (CP6510, Siglent) for current collection. Simply, the packaged non-woven ferroelectret fabrics (dimension of 5 mm by 10 mm) were inserted into porcine tissue for ex vivo testing. As the implantation depth increases, the current shows the same trend as the voltage (**Revised Supplementary Fig. 15c**). Regarding the injection current applied to the nerve, considering that both the sciatic and sacral nerves were located about 1 cm beneath the skin of rats, we therefore studied the current output of ferroelectrets with different sizes at 1cm in pork tissue. We found a positive correlation between the size of ferroelectrets and their current output in the range of 1 mm² to 50 mm² (**Revised Supplementary Fig. 15d**). For the sciatic nerve, the injection current is 13 ± 4 nA owing to the ferroelectret with an area of about 2 mm², while for the sacral visceral nerve, it is 8 ± 4 nA owing to the ferroelectret with an area of about 1 mm².

Revised Supplementary Fig. 15. c, Current generated by the ferroelectrets implanted at 1 cm, 5 cm, and 10 cm under the porcine tissue. **d**, Current outputs as a function of the ferroelectret area from 1 to 50 mm².

References:

1. Kim T, et al. Deep brain stimulation by blood-brain-barrier-crossing piezoelectric nanoparticles generating current and nitric oxide under focused ultrasound. *Nat Biomed Eng* 7, 149-163 (2023).
2. Curry EJ, et al. Biodegradable nanofiber-based piezoelectric transducer. *Proc Natl Acad Sci U S A* 117, 214-220 (2020).
3. Zhang T, et al. Piezoelectric ultrasound energy-harvesting device for deep brain stimulation and analgesia applications. *Science Advances* 8, (2022).
4. Zhu P, Chen Y, Shi J. Piezocatalytic Tumor Therapy by Ultrasound-Triggered and BaTiO₃ - Mediated Piezoelectricity. *Adv Mater* 32, e2001976 (2020).
5. Chen JC, et al. A wireless millimetric magnetolectric implant for the endovascular stimulation of peripheral nerves. *Nature Biomedical Engineering* 6, 706-716 (2022).
6. Herrington TM, Cheng JJ, Eskandar EN. Mechanisms of deep brain stimulation. *J Neurophysiol* 115, 19-38 (2016).
7. Kringelbach ML, Jenkinson N, Owen SL, Aziz TZ. Translational principles of deep brain stimulation. *Nat Rev Neurosci* 8, 623-635 (2007).
8. Chen R, Canales A, Anikeeva P. Neural Recording and Modulation Technologies. *Nat Rev Mater* 2, (2017).
9. Neher E, Sakaba T. Multiple roles of calcium ions in the regulation of neurotransmitter release. *Neuron* 59, 861-872 (2008).

6. Measurement result in Fig. 2b: At what frequency did you pulse the 1 MHz ultrasound transducer? What is the source of large background voltage when the pulse is zero?

Response: We thank the reviewer's comment. We used a mains frequency of 50 Hz to power the 1 MHz ultrasound transducer. It is necessary to point out that the large background of the voltage curve is present in the entire waveform, not just when the ultrasound stops. The source of the background of the voltage curve in **Fig. 2b (left)** may be a variety of signals, such as noise, electromagnetic interference at 50 Hz, and overlapping signals of high-frequency electrical signals at the low sampling rate of the oscilloscope. To confirm the source of the background signal, we simply replaced the ferroelectret film with a non-piezoelectric fibrous polylactic acid film. Subsequently, the same test method as **Fig. 2b** is used for signal collection (**Fig. R5**). We found that the noise and electromagnetic interference signals were much smaller than the background

signals in **Fig. 2b**. Therefore, we can conclude that the background source of the voltage curve is mainly the overlap of the electrical signal of the ferroelectrets. This may be due to the fact that the liquid environment in which the ferroelectret film is located is still in a weak vibration state even if the ultrasound pulse wave stops.

Fig. R5. Voltage output measured in ethanol at a distance of 10 mm from an ultrasound probe to the non-piezoelectric polycaprolactone film, with an ultrasound setup of 1 MHz and 0.5 W cm^{-2} .

7. Since the ultrasound sonication (thereby electrical pulses) was at 1 MHz, what was the underlying mechanism for neuromodulation in in vivo tests/results in Fig. 3 and 4. Also, what is the estimated voltage on the implant in these tests?

Response: Good point. We highly appreciate the reviewer's professional comment. In fact, although many previous studies have successfully used ultrasound with a frequency of 1 MHz for neuromodulation including peripheral nerve stimulation and deep brain stimulation, its exact mechanism is not clearly elucidated which we also have been pondering.¹⁻⁵ A common hypothesis is that neuromodulation electrically stimulates the nervous system by opening voltage-gated ion channels (such as Ca^{2+} , Na^+ , and K^+) in nearby nerve cells and further alters pathological network activity.⁶⁻⁸ *Our study is consistent with this hypothesis that ultrasound-induced electrical pulses promoted the opening of Ca^{2+} channels in SH-SY5Y-derived neuron-like cells (Fig. 2h to k).*

Regarding the estimated voltage applied to neural tissue, we need to take into account the size and depth of the implanted ferroelectret. Therefore, considering that both the sciatic and sacral nerves were located about 1 cm beneath the skin of rats, we studied the voltage output of ferroelectrets with different sizes at 1cm in pork tissue. We found a positive correlation between the size of ferroelectrets and their voltage output in the range of 1 mm^2 to 50 mm^2 (**Revised Supplementary Fig. 15e**). For the sciatic nerve in **Fig. 3**, the estimated voltage is $0.5 \pm 0.1 \text{ V}$ owing to the ferroelectret with an area of about 2 mm^2 . For the sacral splanchnic nerve in **Fig. 4**, the estimated voltage is $0.2 \pm 0.1 \text{ V}$ owing to the ferroelectret with an area of about 1 mm^2 .

Revised Supplementary Fig. 15. e, Voltage outputs as a function of the ferroelectret area from 1 to 50 mm².

References:

1. Kim T, et al. Deep brain stimulation by blood-brain-barrier-crossing piezoelectric nanoparticles generating current and nitric oxide under focused ultrasound. *Nat Biomed Eng* 7, 149-163 (2023).
2. Curry EJ, et al. Biodegradable nanofiber-based piezoelectric transducer. *Proc Natl Acad Sci U S A* 117, 214-220 (2020).
3. Zhang T, et al. Piezoelectric ultrasound energy-harvesting device for deep brain stimulation and analgesia applications. *Science Advances* 8, (2022).
4. Zhu P, Chen Y, Shi J. Piezocatalytic Tumor Therapy by Ultrasound-Triggered and BaTiO(3) - Mediated Piezoelectricity. *Adv Mater* 32, e2001976 (2020).
5. Herrington TM, Cheng JJ, Eskandar EN. Mechanisms of deep brain stimulation. *J Neurophysiol* 115, 19-38 (2016).
6. Kringelbach ML, Jenkinson N, Owen SL, Aziz TZ. Translational principles of deep brain stimulation. *Nat Rev Neurosci* 8, 623-635 (2007).
7. Chen R, Canales A, Anikeeva P. Neural Recording and Modulation Technologies. *Nat Rev Mater* 2, (2017).
8. Neher E, Sakaba T. Multiple roles of calcium ions in the regulation of neurotransmitter release. *Neuron* 59, 861-872 (2008).

Reviewer #3

1. The first observation that drives the decision on this article relates to novelty of the work done. The foundation of the paper revolves around the use of PVDF/ZnO film obtained through electrospinning as the ferroelectret. However, it is well-known that electrospinning can prepare PVDF nanofibers (e.g., *Sensors (Basel)*. 2020 Sep; 20(18): 5214). Incorporating nanoparticles like ZnO is not uncommon in such studies. Additionally, this reviewer has doubts about the claim that their electrospun PVDF/ZnO can be considered a ferroelectret device. Firstly, their voids appear to be very small (Fig. 1d), which, according to calculations and previous experiments by ferroelectret nanogenerators (FENG) researchers, may not generate sufficient piezoelectric effects.

Response: We thank the reviewer's comments. We respectfully disagree with the reviewer at this point, and we appreciate this opportunity to better state the novelty and significance of this paper in the field of piezoelectric materials, soft functional materials, and biomedical devices. Indeed, PVDF nanofibers and the incorporation of nanoparticles including ZnO and BaTO₃ into the fibers have long been reported,^{1,2} but the fact is that their piezoelectricity is still low, limiting their electric physiotherapy as effective receivers unless flexibility is sacrificed. Innovatively, we utilize the cavitation effect to introduce ferroelectret pore structure into the soft electrospun PVDF fiber matrix (please find details from **Supplementary Fig. 1**), which can obtain additional electric dipoles at the surface of the material and inside it, thus enhancing the electrical performance output. *This theory that the electret effect occurs in the gap between ZnO and PVDF is reasonably inferred based on a large amount of experimental data and literature research.* Specifically,

The d_{33} of the ferroelectret fibers is the highest ($56 \pm 2 \text{ pC N}^{-1}$), which is more than twice the air-blowing PVDF fibers ($22 \pm 2 \text{ pC N}^{-1}$) or PVDF/ZnO composite fibers ($25 \pm 2 \text{ pC N}^{-1}$) (**Supplementary Fig. 6**). To understand this enhancement, we studied the polymer chain's orientation and crystallinity. First, we performed thermogravimetric analysis on PVDF fibers, air-blowing fibers, PVDF/ZnO composite fibers, and ferroelectret fibers (**Supplementary Fig. 5a**). All fibers experienced obvious weight loss at 400°C. At this stage, the PVDF molecular chains were decomposed to remove H-F in the molecules. Among them, the decomposition rate of ferroelectret fibers is relatively the lowest, indicating its high crystallinity. Crystallinity was further quantified by 1D XRD (**Supplementary Fig. 5b**). We calculated the crystallinity of the PVDF fibers, air-blowing fibers, PVDF/ZnO composite fibers, and ferroelectret fibers to be 27.5%, 52.1%, 66.3%, and 68.9%, respectively. According to references,³⁻⁵ the enhanced d_{33} can be attributed to the existence of the oriented amorphous fraction that exhibits improved dipole mobility and better chain alignment after the air-blowing process, enhancing the piezoelectric properties of the fiber. Therefore, we used 2D XRD to further study the orientation of the polymer chain in the fiber. As shown in **Supplementary Fig. 5c**, the results reveal that the air-blowing electrospin process can align the polymer chains along the fiber axis direction. The orientation degrees of fibers are quantified using Herman's orientation factor. We calculated the orientation degrees of the PVDF fibers, air-blowing fibers, PVDF/ZnO composite fibers, and ferroelectret fibers, using Herman's orientation factor to be 0.78, 0.83, 0.80, and 0.81, respectively. Overall, the comparable crystallinities between the PVDF/ZnO composite fibers and ferroelectret fibers as well as the comparable orientation degrees between the air-blowing fibers and ferroelectret fibers suggest that the enhanced value of the piezoelectric coefficient is driven neither primarily by a flow induced orientation effect nor by the crystallinity, but rather by another mechanism. Therefore, there must be a synergistic effect from both the PVDF matrix and ZnO particles. We thus

performed TEM characterization on the air-blowing PVDF/ZnO fiber (**Fig. 1d, Supplementary Fig. 4**) and observed cavitation on both sides of the ZnO particles, with the cavities elongated axially along the fiber (i.e., gas blowing direction). The existence of cavities in the vicinity of the ZnO particles, which are found only in the air-blowing fibers and not in the general electrospun composite fibers, leads us to suggest that cavitation in the drawn composite fiber is the major contributor.

From this observation, a mechanism of enhanced piezoelectricity due to the dimensional effect for the air-blowing PVDF/ZnO fiber is proposed (**Fig. 1c, Supplementary Fig. 1**). With initial electrospinning, PVDF undergoes solidification and crystallization. Upon air drawing of the crystallized sample, cavitation takes place around the ZnO particles, forming horizontal pores on the two sides. After electric poling during electrospinning process, ferroelectric domains in PVDF are polarized, generating an electret effect. When deforming the poled PVDF/ZnO fiber during the direct piezoelectric test, the pore volume changes, creating a dimensional effect for enhanced piezoelectricity. That is, the change of dipole density by changing the pore volume induces significantly improved piezoelectricity. Similar PVDF foam electret has been reported to show significantly enhanced piezoelectric performance due to the porous structure.⁶ In this sense, the drawing-aligned oriented amorphous fraction and the increased dielectric constant of the composite have a much weaker contribution to the enhanced d_{33} .^{4,7} It should be noted that conventional foamed ferroelectrets are prone to depolarize and lose their piezoelectric properties rapidly. In this study, for the first time, we have developed a new class of stable PVDF-based ferroelectret devices by creating cavitation using one-step laminar-flow-assisted electrospinning method. This approach paves a novel route towards a new paradigm of fiber-based, high-performance ferroelectret transducers.

In conclusion, the high performance of the thermally drawn PVDF/ZnO fiber can be attributed to cavitation between ZnO particles and the PVDF fiber matrix, and the well-aligned orientated amorphous fraction, which further increases the piezoelectric performance.

As for the reviewer's concern about the small size of the void (200 nm of radial length, 100 nm of axial length), it is unfounded considering the size of the electron itself (diameter $< 10^{-9}$ nm) and its storage. Obviously, our design does not have the problem of too small gaps. For example, as reported in Nature in 2022, Yan et al. developed a ferroelectret composed of a mixture of PVDF-TrFE and BaTiO₃, in which the gap between the BaTiO₃ particles and PVDF-TrFE matrix is about 100-300 nm.⁸ Of course, most of common reports are about foam electrets, and the pore sizes are mostly micron level or even larger.⁹⁻¹² However, large pore sizes can easily depolarize and rapidly lose their piezoelectric properties. Our ferroelectric electrets exhibit stable performance that can last for > 6 months on end (**Revised Supplementary Fig. 13a**). For details, please refer to the response to the second reviewer's comment 3.

Revised Supplementary Fig. 13. a, Comparison of the output voltage before and after the ferroelectret fibers being immersed in a PBS solution at 37 °C for 6 months.

References:

1. Sorayani Bafqi MS, Bagherzadeh R, Latifi M. Fabrication of composite PVDF-ZnO nanofiber mats by electrospinning for energy scavenging application with enhanced efficiency. *Journal of Polymer Research* 22, (2015).
2. Hussein AD, Sabry RS, Abdul Azeez Dakhil O, Bagherzadeh R. Effect of Adding BaTiO₃ to PVDF as Nano Generator. *Journal of Physics: Conference Series* 1294, (2019).
3. Huang Y, et al. Enhanced piezoelectricity from highly polarizable oriented amorphous fractions in biaxially oriented poly(vinylidene fluoride) with pure beta crystals. *Nat Commun* 12, 675 (2021).
4. Zhu Z, et al. Electrostriction-enhanced giant piezoelectricity via relaxor-like secondary crystals in extended-chain ferroelectric polymers. *Matter* 4, 3696-3709 (2021).
5. Li T, et al. High-Performance Poly(vinylidene difluoride)/Dopamine Core/Shell Piezoelectric Nanofiber and Its Application for Biomedical Sensors. *Adv Mater* 33, e2006093 (2021).
6. Zhang Z, Yao C, Yu Y, Hong Z, Zhi M, Wang X. Mesoporous Piezoelectric Polymer Composite Films with Tunable Mechanical Modulus for Harvesting Energy from Liquid Pressure Fluctuation. *Advanced Functional Materials* 26, 6760-6765 (2016).
7. Zhu Z, Rui G, Li R, He H, Zhu L. Effect of Dipole Mobility in Secondary Crystals on Piezoelectricity of a Poly(vinylidene fluoride-co-trifluoroethylene) 52/48 mol % Random Copolymer with an Extended-Chain Crystal Structure. *Macromolecules* 54, 9879-9887 (2021).
8. Yan W, et al. Single fibre enables acoustic fabrics via nanometre-scale vibrations. *Nature* 603, 616-623 (2022).
9. Zhang Y, et al. Ferroelectret materials and devices for energy harvesting applications. *Nano Energy* 57, 118-140 (2019).
10. Bauer S, Gerhard-Multhaupt R, Sessler GM. Ferroelectrets: Soft Electroactive Foams for Transducers. *Physics Today* 57, 37-43 (2004).
11. Li W, Torres D, Wang T, Wang C, Sepúlveda N. Flexible and biocompatible polypropylene ferroelectret nanogenerator (FENG): On the path toward wearable devices powered by human motion. *Nano Energy* 30, 649-657 (2016).
12. Li W, et al. Nanogenerator-based dual-functional and self-powered thin patch loudspeaker or microphone for flexible electronics. *Nature Communications* 8, (2017).

2-1. Secondly, information about the poling process using high voltage was not explained clearly. Even if the authors performed poling, there is no comparison provided between samples with and without high-voltage poling. This reviewer questions whether the enhanced piezoelectric effect is solely due to the intrinsic piezoelectricity of PVDF after the addition of nanoparticles and/or stretching, which has been known to enhance the piezoelectric properties.

Response: We thank the reviewer's comments. It is necessary to point out that we use electrospinning technology to prepare PVDF/ZnO-based fibrous non-woven fabrics. The polarization is carried out simultaneously during the electroinjection process. As described in the 'Methods' section of the original manuscript, 'The electrified liquid droplet was stretched and elongated to fibers from the spinneret via a direct-current, constant-high-voltage (18 kV) power and the as-spun fibers were collected on an aluminum-coated roller collector at 15 cm from the nozzle. '. The calculated polarization electric field is 1.2 kV/cm. To make it easier for the reviewer and more readers to understand our work, we have added the expression of polarization electric field in the manuscript, 'The fiber was stretched and polarized in a high electric field of 1.2 kV/cm' (**Page 3, line 21 in revised manuscript**)

As for the comparison between samples with and without high-voltage polarization, considering that electrospinning cannot proceed without high-voltage electric field, therefore, in this study, we do not and cannot provide unpolarized samples as a comparison group. Of course, to prove the importance of high-voltage polarization, we additionally used the extrusion method as a supplementary experiment (**Fig. R6a**). We set the rotation speed of the roller collection device to 4 mm/s to provide tensile force for the extruded fibers and ensure the formation of pores between ZnO particles and PVDF matrix (**Fig. R6b**). To test the piezoelectric performance of the materials, 10 fibers were arranged side by side and then integrated with top/bottom electrodes. The devices were tested by repeatedly pressing and releasing at a force of ~120 N and low frequency of ~2.5 Hz using an auto step-motor controller (**Fig. R6c**). We found that fibers polarized at 1.2 kV/cm showed significantly improved piezoelectric output (**Fig. R6d**), whether pure PVDF or PVDF/ZnO extruded fibers. It is worth noting that under unpolarized conditions, there is no significant difference between PVDF/ZnO and PVDF, but when polarization voltage is applied, although both improve, PVDF/ZnO improves more obviously. This further illustrates the importance of ferroelectret voids.

For the last comment, as we responded to the reviewer's comment 1, the addition of ZnO nanoparticles and/or stretching contributes weakly to piezoelectric enhancement. To prove that the piezoelectric enhancement mainly originates from the electret effect, we set up PVDF fibers, air-blown PVDF fibers, PVDF/ZnO fibers, and air-blown PVDF/ZnO fibers as experimental groups. The d_{33} of the ferroelectret fibers is the highest ($56 \pm 2 \text{ pC N}^{-1}$), which is more than twice the air-blowing PVDF fibers ($22 \pm 2 \text{ pC N}^{-1}$) or PVDF/ZnO composite fibers ($25 \pm 2 \text{ pC N}^{-1}$) (**Supplementary Fig. 6**). Of course, compared to PVDF fibers ($19 \pm 2 \text{ pC N}^{-1}$), air-blown PVDF fibers and PVDF/ZnO fibers have also improved, but not significantly. Specifically,

We calculated the crystallinity of the PVDF fibers, air-blowing fibers, PVDF/ZnO composite fibers, and ferroelectret fibers to be 27.5%, 52.1%, 66.3%, and 68.9%, respectively (**Supplementary Fig. 5a and b**). The enhanced d_{33} can be attributed to the existence of the oriented amorphous fraction that exhibits improved dipole mobility and better chain alignment after the air-blowing process, enhancing the piezoelectric properties of the fiber. Therefore, we used 2D XRD to further study the orientation of the polymer chain in the fiber (**Supplementary Fig. 5c**). The results reveal that the air-blowing electrospin process can align the polymer chains along the fiber axis direction. The orientation degrees of fibers are quantified using Herman's orientation factor. We calculated the orientation degrees of the PVDF fibers, air-blowing fibers, PVDF/ZnO composite fibers, and ferroelectret fibers, using Herman's orientation factor to be 0.78, 0.83, 0.80,

and 0.81, respectively. Overall, the comparable crystallinities between the PVDF/ZnO composite fibers and ferroelectret fibers as well as the comparable orientation degrees between the air-blowing fibers and ferroelectret fibers suggest that the enhanced value of the piezoelectric coefficient is driven neither primarily by a flow induced orientation effect nor by the crystallinity, but rather by another mechanism. Therefore, there must be a synergistic effect from both the PVDF matrix and ZnO particles. We thus performed TEM characterization on the air-blowing PVDF/ZnO fiber (**Fig. 1d, Supplementary Fig. 4**) and observed cavitation on both sides of the ZnO particles, with the cavities elongated axially along the fiber (i.e., gas blowing direction). The existence of cavities in the vicinity of the ZnO particles, which are found only in the air-blowing fibers and not in the general electrospin composite fibers, leads us to suggest that cavitation in the drawn composite fiber is the major contributor.

From this observation, a mechanism of enhanced piezoelectricity due to the dimensional effect for the air-blowing PVDF/ZnO fiber is proposed (**Fig. 1c, Supplementary Fig. 1**). With initial electrospinning, PVDF undergoes solidification and crystallization. Upon air drawing of the crystallized sample, cavitation takes place around the ZnO particles, forming horizontal pores on the two sides. After electric poling during electrospinning process, ferroelectric domains in PVDF are polarized, generating an electret effect. When deforming the poled PVDF/ZnO fiber during the direct piezoelectric test, the pore volume changes, creating a dimensional effect for enhanced piezoelectricity. *Therefore, we conclude that the electret effect plays a dominant role rather than 'the intrinsic piezoelectricity of PVDF after the addition of nanoparticles and/or stretching'.*

Fig. R6. **a**, Photograph of the device for preparing fibers by hot stretching using the melting method. **b**, Top: Photograph of prepared fibers. Bottom: SEM micrograph and its enlarged view of the fiber cross-section, showing pores formed between ZnO particles and PVDF matrix. **c**, Photograph of the voltage testing device. The piezoelectric performance of the piezodevice is tested by the compression-release experiment by an automotor at ~2.5 Hz operating frequency, where the compressive force can be measured by a commercial pressure sensor during the process.

d, Output voltages from fibers when subjected to repeatedly pressing and releasing at a force of ~120 N and a frequency of ~2.5 Hz.

2-2. The second main observation relates to the discrepancy between the results obtained in this work and prior art. The size of this ferroelectret (5 mm by 5mm) is very unlikely to produce the 6 V mentioned in this article, especially if the target tissue is under human skin. Although it has been found that the response from ferroelectrets increases with frequencies, and get good response in the ultrasonic regions (in published articles that the current manuscript ignores), it is very difficult to understand how such a small device under human skin is capable of responding to ultrasonic frequencies with such large voltage response.

Response: We thank and respect the reviewer's comments. Actually, for inorganic ferroelectric materials, high voltage output over 6 V is common under ultrasonic excitation,^{1,2} with peak-to-peak voltages even reaching up to 80 V.² Also, it is necessary to point out that the peak-to-peak voltage of 6 V is measured at a depth of 10 mm in an ethanol environment, rather than in biological tissues such as subcutaneous tissue. To be precise, in our work, our device (*dimension of 5 mm by 10 mm*) generated a peak-to-peak voltage of approximately 2.5 V at 10 mm below the porcine skin which is similar to human skin (**left, Fig. 2d**). Clearly, the output voltage of the non-woven fabric under the porcine tissue was about 2.5 times lower than in ethanol (*dimension of 5 mm by 5 mm*), owing to the increased acoustic impedance and attenuation in the different media and layered tissue structures. Similarly, it is reported that inorganic ferroelectric materials-based device (6 mm by 6 mm) can still generate a peak-to-peak voltage of approximately 16 V at a depth of 30 mm under porcine tissue.² Obviously, as long as the ultrasound receiver has high performance, it can still respond well to ultrasound even in biological tissues. However, due to the inherent rigidity and biosafety concerns of inorganic materials, they do not meet our application requirements. Our ferroelectret materials exhibit high piezoelectric properties comparable to inorganic materials and excellent biocompatibility, making them more suitable for use in the human body. For details, please refer to the response to the second reviewer's comments 1 and 2.

On the other hand, we respectfully disagree with the reviewer's point of view that '*it has been found that the response from ferroelectrets increases with frequencies*'. For ultrasonic receivers, generally speaking, the output voltage will reach its maximum at the lowest resonant frequency of the ferroelectric materials. Once above or below this frequency, the output voltage will decrease.^{1,3}

Reference:

1. Pop F, Herrera B, Rinaldi M. Lithium Niobate Piezoelectric Micromachined Ultrasonic Transducers for high data-rate intrabody communication. *Nature Communications* 13, (2022).
2. Zhang T, et al. Piezoelectric ultrasound energy-harvesting device for deep brain stimulation and analgesia applications. *Science Advances* 8, (2022).
3. Jiang L, et al. Flexible piezoelectric ultrasonic energy harvester array for bio-implantable wireless generator. *Nano Energy* 56, 216-224 (2019).

2-3. Along this same line, where are the current plots? Did the authors try to characterize output short circuit current from these devices? Did the authors consider the matching impedance between the device and the instrument used to measure the generated voltage? Obtaining real electro-mechanical measurements from ferroelectrets is not a trivial task, and this reviewer believes that

these experiments/studies will generate questions about the electrical voltage output that they are measuring –which could be due to sources other than the ferroelectret.

Response: We thank the reviewer's comment. In the original manuscript, we did not measure short-circuit currents since our goal is to use the local piezopotential yielded from ultrasound receiver to stimulate and regulate nerve tissues, rather than for energy collection or storage. Similarly, since our application aims to stimulate nerves with local piezopotentials, we tend to directly use a voltage probe with 40 M Ω input impedance to collect open-circuit voltage without considering matching impedance, *as described in the 'Methods' section of the original manuscript. (page 13, lines 29 and 30 in revised manuscript)*

Of course, for the sake of rigor, per the reviewer's request, we conducted experiments on current collection and resistance matching. We use a current probe (CP6510, Siglent) for current collection, and the testing method is the same as the voltage collection in **Fig. 2b**. The peak-to-peak short-circuit current is approximately 240 nA (**Fig. R7a**). As shown in **Fig. R7b**, the change of output voltage (blue) and current (red) for the ferroelectret device in an external load resistance range from 100 Ω to 100 M Ω . As the load resistance increases, the output voltage keeps raising until saturation at high resistance (>10 M Ω). On the contrary, the current continuously decreases.

As for the source of voltage, since the entire device is operated in a liquid or biological tissue environment, we can basically exclude the voltage generated by triboelectric effects. This is further verified through experiments on exposed fibrous ferroelectrets in biological tissue fluids (**Fig. 2e and f**). Furthermore, to eliminate electromagnetic and noise interferences, we simply replaced the ferroelectret film with a non-piezoelectric fibrous polylactic acid film. Subsequently, the same test method as **Fig. 2b** is used for signal collection. From **Fig. R5**, we found that the noise and electromagnetic interference signals were much smaller than the voltage signals in **Fig. 2b**. For details, please refer to the response to the second reviewer's comment 6. Therefore, we conclude that the collected voltage mainly comes from the piezoelectric signal of the ferroelectrets.

Fig. R7. a, Current output measured in ethanol at a distance of 10 mm from an US probe to the ferroelectret, with a US setup of 1 MHz and 0.5 W cm $^{-2}$. **b**, The output voltage, and current of the ferroelectret under different load conditions measured in ethanol at 10 mm from an ultrasound probe to the ferroelectret film, with an ultrasound setup of 1 MHz and 0.5 W cm $^{-2}$.

3. Finally, although their application may appear fancy, it takes a quick look to the references and

relevant prior art to realize that this is also not a novel concept –this reviewer has personally read this same application roughly 5 years ago. Lastly, it's disheartening to note that they not only failed to cite any recent relevant work (including a very recent comprehensive review article on ferroelectrets and their applications), but also neglected to reference important studies by other well-known researchers in the field of ferroelectrets.

Response: We thank the reviewer's comment. Regarding the selection of exemplary applications, please allow us to politely elaborate on the reasons why we chose to treat colitis by stimulating the sacral nerve, not for fancy purposes, but based on the shortcomings or deficiencies of the existing treatment protocols. From a rigorous scientific perspective, our application demonstration is based on the fact that the sacral nerve directly innervates the distal colon and rectum.¹ *Based on this, in the past decade, some studies have intermittently reported stimulating the sacral nerve to reduce inflammation and pain in colitis.*²⁻⁵ However, it cannot be ignored that most of these reports inevitably use rigid electrodes, long wires, and external power sources, which are not suitable for long-term and clinical applications. As stated in our original manuscript, these battery-powered tethered systems can restrict natural motions and prevent social interactions.⁷⁻¹⁰ Besides, mechanical mismatch at the rigid electrode–soft neural tissue interface can cause trauma and insertion-related lesions, inflammation reactions, and even neuronal apoptosis, ultimately leading to therapeutic failure.^{7,11,12} We provide a wireless, leadless, and battery-free treatment strategy that precisely regulates and alleviates colitis by triggering stimulation of peripheral nerves with ultrasound pulses. Besides, our device exhibits high mechanical conformability to dynamic motion nerve tissue and can wrap around the target peripheral nerve without affecting normal development and movement. Of course, our study of sacral nerve stimulation for colitis is just a demonstration of its application, which can provide a solid foundation for electromodulation of peripheral nerves to treat disease.

As for the comment we did not cite the latest ferroelectret-related research, we believe it is unfounded. In fact, we have conducted extensive analysis and citation of existing electrets, especially those based on PVDF piezoelectric materials, in the original manuscript (**Revised Supplementary Table 1**). As is well known, depending on the charge carrier type, electrets are divided into charge electrets and dipole electrets. The electret charges are either real excess charges (charge electrets) or result from oriented dipoles (dipole electrets).¹³ Ferroelectrets are a member of the electret family (charge electrets) based on nonpolar polymers with a porous foam structure with open or closed cells where the internal surfaces carry positive and negative charges. Foamed ferroelectrets have been widely fabricated based on polyethylene and polypropylene polymers because of their good insulating properties. However, their poor stability limits their application as they are prone to depolarize and lose their piezoelectric properties rapidly.¹⁴ Notably, as a representative of semicrystalline dipole electrets, PVDF was found to exhibit considerable piezoelectric and charge stability as early as the 1960s and early 1970s.^{15,16} Therefore, in this study, for the first time, we introduced ferroelectret pore structure by using a one-step laminar-flow-assisted electrospinning method, further increasing the number of electric dipoles in PVDF dipole electrets, and thus developing a new class of stable fibrous PVDF-based ferroelectret devices. This approach paves a novel route towards a new paradigm of high-performance and stable soft ferroelectret transducers. For details, please refer to the description of the **Supplementary Fig. 6** in Supplementary Materials. (**pages S8 to S10 in Supplementary Materials**)

Reference:

1. Furness JB. The enteric nervous system and neurogastroenterology. *Nat Rev Gastroenterol Hepatol* 9, 286-294 (2012).
2. Bregeon J, et al. Improvement of Refractory Ulcerative Proctitis With Sacral Nerve Stimulation. *J Clin Gastroenterol* 49, 853-857 (2015).
3. Nunes NS, et al. Therapeutic ultrasound attenuates DSS-induced colitis through the cholinergic anti-inflammatory pathway. *EBioMedicine* 45, 495-510 (2019).
4. Guo J, Jin H, Shi Z, Yin J, Pasricha T, Chen JDZ. Sacral nerve stimulation improves colonic inflammation mediated by autonomic-inflammatory cytokine mechanism in rats. *Neurogastroenterol Motil* 31, e13676 (2019).
5. Tu L, Gharibani P, Zhang N, Yin J, Chen JD. Anti-inflammatory effects of sacral nerve stimulation: a novel spinal afferent and vagal efferent pathway. *Am J Physiol Gastrointest Liver Physiol* 318, G624-G634 (2020).
6. Mohebbi, A., Mighri, F., Aji, A. & Rodrigue, D. Cellular Polymer Ferroelectret: A Review on Their Development and Their Piezoelectric Properties. *Adv. Polym. Technol.* 37, 468–483 (2018).
7. Won SM, Cai L, Gutruf P, Rogers JA. Wireless and battery-free technologies for neuroengineering. *Nat Biomed Eng*, (2021).
8. Burton A, et al. Wireless, battery-free subdermally implantable photometry systems for chronic recording of neural dynamics. *Proc Natl Acad Sci U S A* 117, 2835-2845 (2020).
9. Wei Z, et al. Physical Cue - Based Strategies on Peripheral Nerve Regeneration. *Advanced Functional Materials*, (2022).
10. Jin F, et al. Biofeedback electrostimulation for bionic and long-lasting neural modulation. *Nat Commun* 13, 5302 (2022).
11. Kavvadias T, Huebner M, Brucker SY, Reisenauer C. Management of device-related complications after sacral neuromodulation for lower urinary tract disorders in women: a single center experience. *Arch Gynecol Obstet* 295, 951-957 (2017).
12. del Valle J, Navarro X. Interfaces with the peripheral nerve for the control of neuroprostheses. *Int Rev Neurobiol* 109, 63-83 (2013).
13. Graz I, Mellinger A. Polymer Electrets and Ferroelectrets as EAPs: Fundamentals. In: *Electromechanically Active Polymers: A Concise Reference* (ed Carpi F). Springer International Publishing (2016).
14. Mohebbi A, Mighri F, Aji A, Rodrigue D. Cellular Polymer Ferroelectret: A Review on Their Development and Their Piezoelectric Properties. *Advances in Polymer Technology* 37, 468-483 (2016).
15. Kawai H. The Piezoelectricity of Poly (vinylidene Fluoride). *Japanese Journal of Applied Physics* 8, (1969).
16. McFee JH, Bergman JG, Crane GR. Pyroelectric and Nonlinear Optical Properties of Poled Polyvinylidene Fluoride Films. *IEEE Transactions on Sonics and Ultrasonics* 19, 305-314 (1972).

REVIEWERS' COMMENTS:

Reviewer #1 (Remarks to the Author):

This work is interesting, and I recommend it to be published. But a minor revision is needed.

- 1、 The PVDF and ZnO composite exhibits excellent piezoelectric performance. Could you add more details about its electrical output under ultrasound with different frequency ?
- 2、 As the authors described, the ultrasound transducer is unfocused, and its aperture is much larger than the implanted device, which leads to that some ultrasound will act on the nerve. How to prove the nerve is stimulated by electrical signal and not modulated by ultrasound directly?

Reviewer #2 (Remarks to the Author):

Thank you for addressing all my questions.

Reviewer #3 (Remarks to the Author):

The authors have done a good job at addressing the concerns raised in the first round of reviews. I still believe their summary of prior art can be improved by including some of the discussions presented in the rebuttal letter in the actual manuscript, especially regarding the FENG results. However, I do not think this should be considered a major obstacle for publishing the work. I believe the work is publishable in the current form, after including some of the critical responses into the manuscript.

REVIEWERS' COMMENTS:

Reviewer #1 (Remarks to the Author):

This work is interesting, and I recommend it to be published. But a minor revision is needed.

- 1、 The PVDF and ZnO composite exhibits excellent piezoelectric performance. Could you add more details about its electrical output under ultrasound with different frequency?
- 2、 As the authors described, the ultrasound transducer is unfocused, and its aperture is much larger than the implanted device, which leads to that some ultrasound will act on the nerve. How to prove the nerve is stimulated by electrical signal and not modulated by ultrasound directly?

Reviewer #2 (Remarks to the Author):

Thank you for addressing all my questions.

Reviewer #3 (Remarks to the Author):

The authors have done a good job at addressing the concerns raised in the first round of reviews. I still believe their summary of prior art can be improved by including some of the discussions presented in the rebuttal letter in the actual manuscript, especially regarding the FENG results. However, I do not think this should be considered a major obstacle for publishing the work. I believe the work is publishable in the current form, after including some of the critical responses into the manuscript.

Reviewer #1

This work is interesting, and I recommend it to be published. But a minor revision is needed.

Response: We appreciate the reviewer's high recognition and constructive suggestions.

1、 The PVDF and ZnO composite exhibits excellent piezoelectric performance. Could you add more details about its electrical output under ultrasound with different frequency?

Response: Per the reviewer's request, we tested and counted the voltage output of ferroelectrets driven by ultrasound at frequencies ranging from 200 kHz to 1400 kHz (**Fig. R1**). It can be inferred that the first-order resonant frequency of our fibrous ferroelectret is around 700 kHz due to the high voltage output at this frequency. Thanks to the high performance of our ferroelectric materials, it is ensured that the ultrasound receiver can generate sufficient electrical stimulation pulses at non-resonant frequencies. Of course, we still appreciate the constructive questions raised by the reviewers, which motivate us to further tailor electromechanical coupling materials that resonate with high-frequency ultrasound (>1 MHz) in future studies to improve piezoelectric properties and biocompatibility.

Fig. R1. Normalized voltage output of ferroelectrets driven by different ultrasonic frequencies.

2、 As the authors described, the ultrasound transducer is unfocused, and its aperture is much larger than the implanted device, which leads to that some ultrasound will act on the nerve. How to prove the nerve is stimulated by electrical signal and not modulated by ultrasound directly?

Response: Good point. In our study, we conducted comprehensive and rigorous group experiments and demonstrated that nerves are stimulated by ultrasound-induced electrical pulses rather than directly modulated by ultrasound. Specifically, first, we performed in vitro Ca^{2+} channels activation experiments. Although ultrasonic stimulation alone also caused an influx of

calcium ions, it was almost negligible compared with piezoelectric materials including blown PVDF fibers, PVDF/ZnO composite fibers, and ferroelectric electret fibers (**Fig. 2k, Supplementary Fig. 16**). Secondly, from the perspective of the response of the gastrocnemius muscle and the therapeutic effect of colitis in vivo, compared with the control groups including ultrasound alone and sham groups, the ultrasound-induced electric stimulation shows significant advantages (**Fig. 3j to l, Fig. 4**). Therefore, from these data, we can conclude that ultrasound-induced electric stimulation is the main factor for the neuromodulation.

Reviewer #2

Thank you for addressing all my questions.

Response: We are very grateful to the reviewer's high recognition. The reviewer's professional comments and suggestions help us to improve the quality of the manuscript a lot. Thanks again!

Reviewer #3

The authors have done a good job at addressing the concerns raised in the first round of reviews. I still believe their summary of prior art can be improved by including some of the discussions presented in the rebuttal letter in the actual manuscript, especially regarding the FENG results. However, I do not think this should be considered a major obstacle for publishing the work. I believe the work is publishable in the current form, after including some of the critical responses into the manuscript.

Response: We greatly respect and appreciate the reviewer's comments and suggestions. Regarding ferroelectret nanogenerators (FENG), we have elaborated extensively in the article, especially in the Supplementary Information, including conventional foamed ferroelectrets, PVDF foam electrets and semi-crystalline dipole electrets represented by PVDF. These discussions further improve the system of our study and better allow readers to understand the innovation of our work.